https://doi.org/10.1038/s41467-019-09734-5　　**OPEN**

# AURKB as a target in non-small cell lung cancer with acquired resistance to anti-EGFR therapy

Jordi Bertran-Alamillo [1], Valérie Cattan[2], Marie Schoumacher [2], Jordi Codony-Servat[1],
Ana Giménez-Capitán[1], Frédérique Cantero[2], Mike Burbridge[2], Sonia Rodríguez[1], Cristina Teixidó[1,8],
Ruth Roman[1], Josep Castellví [1], Silvia García-Román[1], Carles Codony-Servat[1], Santiago Viteri[3],
Andrés-Felipe Cardona[4,5], Niki Karachaliou[3], Rafael Rosell[1,3,6,7] & Miguel-Angel Molina-Vila[1]

Non-small cell lung cancer (NSCLC) tumors harboring mutations in *EGFR* ultimately relapse
to therapy with EGFR tyrosine kinase inhibitors (EGFR TKIs). Here, we show that resistant
cells without the p.T790M or other acquired mutations are sensitive to the Aurora B
(AURKB) inhibitors barasertib and S49076. Phospho-histone H3 (pH3), a major product of
AURKB, is increased in most resistant cells and treatment with AURKB inhibitors reduces the
levels of pH3, triggering G1/S arrest and polyploidy. Senescence is subsequently induced in
cells with acquired mutations while, in their absence, polyploidy is followed by cell death.
Finally, in NSCLC patients, pH3 levels are increased after progression on EGFR TKIs and high
pH3 baseline correlates with shorter survival. Our results reveal that AURKB activation is
associated with acquired resistance to EGFR TKIs, and that AURKB constitutes a potential
target in NSCLC progressing to anti-EGFR therapy and not carrying resistance mutations.

[1] Laboratory of Oncology, Pangaea Oncology, Quiron Dexeus University Hospital, 08028 Barcelona, Spain. [2] Institut de Recherches Internationales Servier,
92284 Suresnes, France. [3] Instituto Oncológico Dr. Rosell, Quiron-Dexeus University Hospital, 08028 Barcelona, Spain. [4] Clinical and Translational Oncology
Group, Institute of Oncology, Fundacion Santa Fe de Bogotá, Bogotá 110111, Colombia. [5] Foundation for Clinical and Applied Cancer Research (FICMAC),
Bogotá 110111, Colombia. [6] Catalan Institute of Oncology, Hospital Germans Trias i Pujol, 08916 Badalona, Spain. [7] Germans Trias i Pujol, Health Sciences
Institute and Hospital, Campus Can Ruti, 08916 Badalona, Spain. [8] Present address: Servicio Anatomía Patológica, Hospital Clínic de Barcelona, Barcelona
08036, Spain. Correspondence and requests for materials should be addressed to M.-A.M-V. (email: mamolina@panoncology.com)

Lung tumors are the leading cause of cancer mortality worldwide[1], being non-small cell lung cancer (NSCLC) the most common subtype. Activating mutations in the Epidermal Growth Factor Receptor (EGFR) gene are present in 14–17% of advanced NSCLC in European populations[2] and 40–50% in East Asia[3]. Tumors with the most common alterations, exon 19 deletions and exon 21 point mutations (p.L858R), are sensitive to EGFR tyrosine kinase inhibitors (EGFR TKIs) such as gefitinib[4], erlotinib[5], or afatinib[6]. However, treated patients ultimately develop resistance, leading to disease progression[5].

The most frequent mechanism associated with acquisition of resistance is the p.T790M mutation in EGFR[7], which emerges in 40–60% of tumors progressing on first and second generation EGFR TKIs. Additional mechanisms include amplification, overexpression and/or autocrine activation loops involving membrane receptors such as MET proto-oncogene (MET), Fibroblast Growth Factor Receptor (FGFR) or AXL receptor tyrosine kinase (AXL)[8–11]. Finally, activating mutations of BRAF and PIK3CA, amplification of MAPK1, gene fusions involving FGFR3, histological transformation, and epithelial-to-mesenchymal transition (EMT) have also been associated with progression to EGFR TKIs[8].

The characterization of the mechanisms responsible for acquired resistance to EGFR TKIs is of both fundamental and clinical interest and drugs that target some of the proteins mentioned above are currently in clinical trials. Regarding the p.T790M, third generation EGFR TKIs are active against it and one of them, osimertinib, has been approved for second line treatment[12]. However, similarly to gefitinib, erlotinib or afatinib, patients ultimately progress. Mechanisms of resistance to third generation EGFR TKIs include loss of the p.T790M mutation[13], emergence of a further mutation in exon 20 of EGFR (p.C797S)[14], MET and HER2 activation, and de novo mutations in KRAS[15].

In order to gain insight into the mechanisms associated with acquired resistance in EGFR-mut NSCLC, we have generated cell lines resistant to EGFR TKIs. Unexpectedly barasertib, an Aurora B (AURKB) inhibitor[16,17], shows antitumor activity in resistant cells lacking the p.T790M or other acquired mutations. AURKB is a ubiquitously expressed serine/threonine kinase that phosphorylates histone H3 on Ser10 and variant centrosome protein A on Ser7 in early G2, leading to condensation of chromatin[18]. Later, it regulates the spindle assembly complex, and inhibition or loss of AURKB leads to defective chromosome segregation and polyploidy[19]. Amplification or overexpression of AURKB has been associated with poor prognosis in several human tumors and AURKB inhibitors are in phase I–II clinical trials for leukemia[18,20]. AURKB has also been implicated in resistance to certain antitumor agents, such as aromatase inhibitors in breast carcinoma[21], paclitaxel in NSCLC[22], cetuximab in head and neck squamous cell carcinoma[23], or vemurafenib in melanoma[24]. However, no role has been reported for AURKB in the context of resistance to targeted therapies in NSCLC.

Our results indicate that AURKB is activated in NSCLC tumor cells with acquired resistance to EGFR TKIs and can be a therapeutic target in absence of resistance mutations. Clinical trials are thus warranted to determine the efficacy of multi-targeted agents inhibiting not only RTKs, but also AURKB, in EGFR-mutant patients progressing on EGFR TKIs and not harboring acquired mutations.

## Results

**Resistance to EGFR TKIs in PC9 cells is associated with different alterations**. We had previously generated six EGFR-TKI-resistant cell lines by treating EGFR-mut, TKI-sensitive PC9 cells with increasing concentrations of gefitinib (PC9-GR1–5) or erlotinib (PC9-ER). While all six of them retained the 15-bp deletion in the EGFR gene present in the parental PC9, the p.T790M mutation only emerged in PC9-GR1 and GR4[25]. Both cell lines were sensitive to osimertinib (Table 1). Subsequently, we generated 17 additional lines resistant to osimertinib by treating PC9-GR1 and GR4 with increasing concentrations of the drug; eight of them lost the p.T790M mutation and five also the exon 19 deletion. The p.C797S mutation did not emerge in any case. Six of the osimertinib-resistant cell lines were selected for further work, together with the six lines resistant to first generation EGFR TKIs (Fig. 1a and Table 1). Next generation sequencing (NGS) did not reveal other acquired mutations in BRAF, KRAS, NRAS, or EGFR. The PC9-GR1 and GR4 cell lines, harboring the p.T790M mutation at 25 and 38% allelic fractions, were similar to the parental PC9 under microscopic examination; the rest of the lines showed a more mesenchymal phenotype (Supplementary Fig. 1a).

Western blotting, immunohistochemistry (IHC) and mRNA analyses revealed overexpression of AXL in 10 of the resistant cell lines, MET activation in three and FGFR1 upregulation in two (Figs. 1b–c, Table 1 and Supplementary Fig. 1b). HER2 and MET were not amplified by FISH or NGS in any case. Molecular alterations frequently co-occurred (Table 1). Interestingly, GAS6 expression was significantly elevated in all the resistant cells, particularly in those with AXL upregulation (Fig. 1d and Supplementary Fig. 1c).

**Resistant cells are insensitive to AXL, MET, or FGFR1 inhibition**. Next, we used viability assays to determine the sensitivity of the PC9-derived cell lines to several targeted agents (Table 1). As expected, p.T790M-negative cells resistant to first generation EGFR TKIs (PC9-GR2, GR3, GR5, and ER) were insensitive to afatinib and osimertinib, in contrast to the p.T790M-positive cells (PC9-GR1 and GR4). The osimertinib-resistant lines derived from PC9-GR1 and GR4 also acquired resistance to afatinib and remained insensitive to first generation EGFR TKIs.

The resistant cell lines with AXL upregulation had IC50s around 2–3 μM for the AXL inhibitor BGB324, indistinguishable from the parental PC9 or from the resistant cells not over-expressing AXL. A similar behavior was observed in the case of the MET inhibitors capmatinib and crizotinib, where the IC50s did not correlate with MET activation. Resistant cells also remained largely insensitive to the combination of BGB324 with capmatinib (Supplementary Fig. 2). The FGFR1 over-expressing PC9-GR5 cells showed an IC50 of 2.3 μM for the FGFR1 inhibitor nintedanib; only 2–10 times lower than the rest of the panel.

Western blotting showed that crizotinib at 2 μM effectively suppressed the phosphorylation of MET in PC9-GR1, while BGB324 at the same concentration inhibited the activation of AXL in PC9-ER, and nintedanib the phosphorylation of FGFR substrate 2 (FRS2), the main downstream effector of FGFR1, in PC9-GR5. These results demonstrated that these TKIs, despite showing limited antiproliferative activity in the resistant cells, were able to block their RTK targets at the concentrations used in the MTT assays (Supplementary Fig. 3a).

Finally, since upregulation of AXL was common in our panel of resistant cell lines, we silenced AXL expression in two non-p.T790M cells, PC9-ER, and PC9-GR3. The AXL-silenced clones did not show any morphological changes and had growth curves indistinguishable from the control clones. In addition, no significant differences were found in the sensitivity to any of the drugs tested, including gefitinib (Supplementary Table 1 and Supplementary Fig. 3b–d).

**Table 1 Characteristics (left) and IC50s (μM, right) for selected drugs of the PC9-derived cell lines used in the study**

| Cell line | Doublingt time (h) | EGFR Del19 | EGFR 790 M | AXL over-expression | MET activation | FGFR1 over-expression | Gefitinib (EGFR) | Erlotinib (EGFR) | Afatinib (EGFR) | Osimertinib (EGFR) | S49076 (AXL, MET, FGFR1, AURKB) | Foretinib (MET, KDR, AXL, AURKB) | Crizotinib (MET, ALK) | Capmatinib (MET) | BGB324 (AXL) | Nintedanib (FGFR) |
|---|---|---|---|---|---|---|---|---|---|---|---|---|---|---|---|---|
| PC9 | 24 ± 2 | + | v− | − | − | − | 0.04 | 0.005 | 0.003 | 0.05 | >50 | 1.6 | 2.0 | >25 | 2.3 | 11.4 |
| PC9-GR1 | 27 ± 3 | + | + | − | + | − | 12.2 | 4.8 | 0.4 | 0.03 | >50 | 1.5 | 2.4 | >25 | 2.3 | 14.3 |
| PC9-GR2 | 31 ± 1 | + | − | + | + | − | 14.9 | 25.0 | 3.8 | 3.0 | 1.2 | 0.6 | 2.3 | >25 | 2.5 | 5.7 |
| PC9-GR3 | 30 ± 1 | + | − | + | − | − | 15.8 | 33.7 | 4.8 | 5.5 | 0.3 | 0.5 | 1.0 | >25 | 2.7 | 8.1 |
| PC9-GR4 | 29 ± 1 | + | + | + | − | − | 6.5 | 3.8 | 0.4 | 0.02 | >50 | 3.6 | 2.1 | >25 | 3.3 | 12.5 |
| PC9-GR5 | 31 ± 5 | + | − | ++ | − | ++ | 18.1 | 21.8 | 3.6 | 5.3 | 0.9 | 0.6 | 2.1 | 19.8 | 2.7 | 2.3 |
| PC9-ER | 27 ± 2 | + | − | ++ | − | − | 12.2 | 28.6 | 6.8 | 3.2 | 0.3 | 0.7 | 1.4 | >25 | 3.3 | 7.8 |
| PC9-GR1-AZD1 | 32 ± 5 | − | − | ++ | − | − | 21.0 | 43.9 | 15.3 | 8.3 | 0.2 | 0.6 | 0.7 | >25 | 2.1 | 24.2 |
| PC9-GR1-AZD2 | 25 ± 4 | − | − | ++ | − | − | 20.0 | 49.8 | 17.0 | 8.8 | 0.3 | 0.7 | 1.3 | >25 | 2.4 | 13.3 |
| PC9-GR1-AZD3 | 26 ± 6 | + | − | + | + | − | 18.7 | 42.4 | 15.4 | 5.7 | 0.2 | 0.6 | 0.6 | >25 | 2.1 | 24.9 |
| PC9-GR1-AZD4 | 30 ± 3 | − | − | + | − | − | 18.2 | 35.5 | 13.5 | 6.4 | 0.3 | 0.7 | 0.7 | >25 | 2.1 | 14.3 |
| PC9-GR4-AZD1 | 26 ± 4 | + | + | + | − | − | 10.5 | 43.2 | 7.1 | 5.1 | >50 | 2.5 | 1.9 | 21.9 | 2.6 | 13.7 |
| PC9-GR4-AZD2 | 34 ± 7 | + | +[a] | − | − | + | 13.7 | 41.2 | 2.6 | 7.5 | 0.4 | 4.2 | 5.8 | >25 | 2.3 | 22.2 |
| H1975 | 21 ± 1 | − | + | − | nd | − | 9.1 | 24.5 | 0.0005 | 0.23 | 22.1 | 2.5 | 1.5 | nd | 1.8 | nd |

The H1975 cell line is also included in the table. The targets of the drugs are indicated between parentheses. Doubling times are expressed as means ± SD of at least three different determinations
[a]Allelic fraction 0.03%

**p.T790M-negative cells are sensitive to AURKB inhibition.** Of all the TKIs tested foretinib, a type II inhibitor that targets MET, AXL, and other RTKs but also AURKB[26], showed the lowest IC50 in most of the resistant cell lines, particularly in those p.T790M-negative (Table 1). This result prompted us to test other AURKB inhibitors. We found that barasertib, a drug specifically targeting AURKB, had a strong antiproliferative effect in MTT assays in p. T790M-negative cells, with IC50s < 0.06 μM in contrast to >10 μM in the p.T790M-positive and the parental PC9 cells (Table 2, Fig. 1e, Supplementary Fig. 4). The only exception was PC9-GR2, a p.T790M-negative cell line with MET activation, where barasertib showed little effect. The AURKA/AURKB inhibitor tozasertib was also very active against non-p.T790M cells, although the differences in the IC50s were less pronounced than in the case of barasertib. In view of these results, we included in our study the H1975 NSCLC cell line, a model of intrinsic resistance to EGFR TKIs that harbors a p.L858R and a p.T790M mutation in *EGFR* at 75% allelic fraction (Table 1). Similarly to the p.T790M-positive PC9-derived cells, H1975 was found to be resistant to barasertib and tozasertib (Table 2, Fig. 1e).

Cells treated with barasertib, particularly those p.T790M-positive, appeared significantly enlarged, vacuolized and often multinucleated under microscopic examination (Supplementary Fig. 5a). In consequence, growth curves were determined for PC9-ER (p.T790M negative) and PC9-GR4 (p.T790M positive) cells comparing direct counting vs. viability assays (MTT). Remarkable differences were observed in the case of the PC9-GR4, with a significant dose-dependent effect of AURKB inhibition on cell numbers, which was not reflected in total cell metabolism, as estimated by MTT, probably as a result of the considerably larger size of the cells (Supplementary Fig. 5b-c).

**S49076 is active against p.T790M-negative resistant cells.** Next, we tested the effects of S49076, another type II inhibitor, in our panel of resistant cells. S49076 targets MET, AXL, and FGFR1/2/3[27] but also shows a significant in vitro activity against AURKB, with an IC50 of 3 nM on the isolated enzyme[28]. Similarly to barasertib or tozasertib, parental PC9 and the PC9-derived cell lines carrying the p.T790M were resistant to the S49076, with IC50s over 50 μM, while the p.T790M-negative cell lines were sensitive to the drug (Supplementary Fig. 6a). Interestingly, the PC9-GR4-AZD2 cells, which harbor the p.T790M mutation at very low allelic fractions, showed an intermediate behavior. Although barasertib had lower IC50s than S49076 in most of the p.T790M-negative cells, surviving populations were often observed at ≥1 μM concentrations of barasertib but not of S49076 (Table 2).

S49076 was able to block the phosphorylation of its target RTKs at concentrations within the range used in the MTT assays (Supplementary Fig. 3a). However, several lines of evidence indicated that inhibition of RTKs was insufficient to explain the antitumor effects of S49076 and that AURKB inhibition was playing a significant role. First, S49076 showed significantly stronger antiproliferative activity than the combination of an AXL and a MET inhibitor (Supplementary Fig. 2). Second, the *AXL*-silenced cells showed IC50s for S49076 indistinguishable from the parental or control cells (Supplementary Table 1 and Supplementary Fig. 3d). Third, western blotting revealed that S49076 induced only a moderate reduction in the levels of pAKT and/or pERK1/2 in the non-p.T790M-resistant cells (Supplementary Fig. 6b-c). Fourth, non-p.T790M-resistant cells treated with S49076 showed morphological changes similar to those observed in the case of barasertib and remarkable differences between direct counting vs. viability assays (MTT) (Supplementary Fig. 7).

**pH3 but not AURKB is elevated in resistant cells.** The previous results suggested that AURKB could be a target in EGFR-TKI-resistant cells negative for the p.T790M mutation. A series of experiments were performed to test this hypothesis. First, we determined the levels of total AURKB by mRNA expression analyses, Western blotting and immunofluorescent staining, and found no significant differences between the parental and the resistant PC9 cells (Figs. 2a–c). In contrast, a significant increase in pH3, the main product of AURKB activity, was observed in eight of the 12 EGFR-TKI-resistant cells, both p.T790M-positive and negative (Figs. 2d–e). During cell cycle progression, histone H3 is phosphorylated in early G2 and binds to mitotic chromosomes. However, the total increase of pH3 levels in the resistant cell lines did not associate with a higher percentage of proliferating cells, as revealed by Ki67 IHC (Figs. 2f–g).

IHC also showed that the pH3 positivity in the parental PC9 line was mainly circumscribed to mitotic nuclei or dividing cells, which showed an intense staining (3+). In contrast, in the case of the EGFR-TKI-resistant cells, a significant number of apparently resting cells also showed a weak or moderate nuclear staining (1+ or 2+). Interestingly, the H-score for mitotic (3+) pH3 closely correlated with Ki67 immunostaining in our panel of resistant lines, but the H-score corresponding to apparently resting cells (1+ or 2+) did not (Fig. 2g). These results were coincident with immunofluoresence studies, where pH3 staining

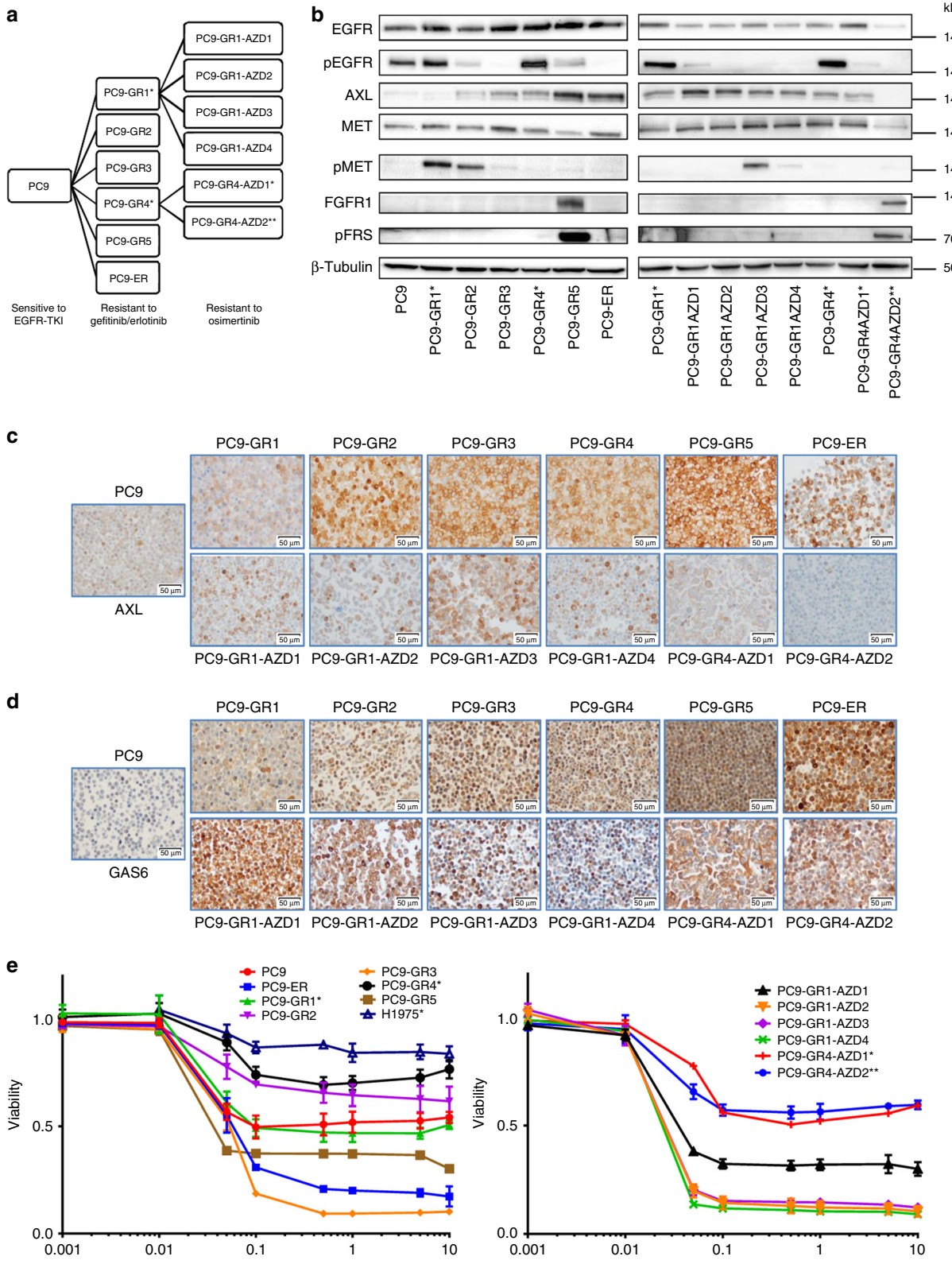

**Fig. 1** Characterization of a panel of PC9-derived cell lines resistant to EGFR TKIs. **a** Diagram summarizing the establishment of the panel. Asterisks indicate cell lines harboring the p.T790M-resistant mutation at >0.1% (*) or < 0.1% (**) allelic fraction. **b** Western blot analysis of proteins and phospho-proteins related with the emergence of resistance. pFRS, phospho-FGFR substrate 2. **c** IHC of AXL. **d** IHC of GAS6. Scale bars indicate 50 μm. **e** Dose-response curves of the parental, the EGFR-TKI-resistant and the H1975 cells to the AURKB inhibitor barasertib at 72 h. Values shown are means ± standard deviation (SD), experiments were conducted in tri (n = 3) or quadruplicates (n = 4). In each experiment, every concentration of drug was tested in sextuplicates (n = 6)

**Table 2 Sensitivity to Aurora inhibitors of the PC9-derived cell lines used in the study**

| Cell line[a] | Barasertib (AURKB) IC50 (µM) | Barasertib (AURKB) % Cells | Tozasertib (AURKA, AURKB) IC50 (µM) | Tozasertib (AURKA, AURKB) % Cells | S49076 (AURKB, AXL, MET, FGFR1) IC50 (µM) | S49076 (AURKB, AXL, MET, FGFR1) % Cells |
|---|---|---|---|---|---|---|
| PC9 | 12.7 | 52 | 0.7 | 44 | >50 | 55 |
| PC9-GR1 | 13.9 | 45 | 5.7 | 65 | >50 | 56 |
| PC9-GR2 | 30.7 | 69 | 0.5 | 47 | 1.2 | 54 |
| PC9-GR3 | 0.05 | 9 | 0.1 | 14 | 0.3 | 4 |
| PC9-GR4 | 34.4 | 67 | 7.8 | 63 | >50 | 65 |
| PC9-GR5 | 0.04 | 37 | 0.2 | 25 | 0.9 | 33 |
| PC9-ER | 0.06 | 20 | 0.2 | 35 | 0.3 | 3 |
| PC9-GR1-AZD1 | 0.04 | 32 | 0.2 | 37 | 0.2 | 9 |
| PC9-GR1-AZD2 | 0.03 | 12 | 0.05 | 14 | 0.3 | 8 |
| PC9-GR1-AZD3 | 0.03 | 15 | 0.08 | 26 | 0.2 | 6 |
| PC9-GR1-AZD4 | 0.03 | 10 | 0.1 | 16 | 0.3 | 7 |
| PC9-GR4-AZD1 | 27.5 | 64 | 2.6 | 53 | >50 | 57 |
| PC9-GR4-AZD2 | 23.2 | 55 | 4.6 | 50 | 0.4 | 31 |
| H1975 | >50 | 83 | 28.8 | 84 | 22.1 | 76 |

The H1975 cell line is also included. The targets of the drugs are indicated between parentheses. The IC50s and percentage of surviving cells at 1 µM inhibitor (in MTT assays) are indicated
[a]In italics, cells carrying the T790M mutation at allelic fractions >1%

in resistant cells was present in dividing but also in apparently resting nuclei, often concentrated in foci (Fig. 3).

Western blotting and immunofluorescence demonstrated a dose-dependent reduction in pH3 by barasertib at concentrations ≥0.05 µM in the p.T790M-negative PC9-ER and the p.T790M-positive PC9-GR4 cells. Similar effects were observed in the case of S49076 at ≥0.1 µM (Figs. 3a–b, Supplementary Fig. 6d–e). The reduction in pH3 levels was primarily observed in apparently resting cells, with weakening or disappearance of foci, while the strong pH3 staining in mitotic cells was not altered (Fig. 3b).

**AURKB silencing de-sensitizes non-p.T790M cells to AURKB inhibition.** Next, we transfected PC9-ER cells with shRNA lentivirus particles to silence *AURKB*. The number of puromycin-resistant clones was low and they took several weeks to emerge. Western blotting, immunofluorescence and gene expression analyses confirmed partial, but not complete, silencing of *AURKB* (Figs. 3c–d). Although the doubling time of the clones with *AURKB* silencing and the parental PC9-ER cells was not significantly different (Supplementary Table 1), microscopic observation revealed profound changes in the morphology of the partly silenced cells, which uniformly showed an enlarged cytoplasm, aberrant mitotic figures and a significant fraction of polynucleated cells (Supplementary Fig. 8). Finally, immunofluorescence demonstrated reduced levels of pH3 in apparently resting but not in dividing cells associated with *AURKB* partial silencing (Fig. 3d). These changes in pH3 staining strongly resembled those observed when treating parental PC9-ER and PC9-GR4 cells with AURKB inhibitors.

Then, we examined the sensitivity to several drugs of the PC9-ER clones partially silenced for *AURKB*. No significant differences were observed in the case of gefitinib, BGB324 or other TKIs. In contrast, the partly silenced clones showed a >100-fold increase in the IC50s for barasertib and S49076, compared to parental or control-transfected PC9-ER cells (Fig. 3e and Supplementary Table 1), confirming that AURKB is the main target of both drugs in PC9-ER. This result was further validated using CRISPR/Cas9 system to knockout *AURKB* from PC9-ER. Similarly to the results obtained when using shRNA, we could only isolate clones with partial silencing (Supplementary Fig. 8b). Also, the *AURKB* silenced clones showed a >10-fold increase in the IC50 for barasertib (Supplementary Fig. 8c), while the IC50s for gefitinib remained unchanged.

**S49076 has antitumor activity on PC9-ER xenografts.** Animal experiments revealed that S49076 inhibited the growth of PC9-ER xenografts in a dose-dependent way. At the end of the 3-week treatment, there was a 55% reduction in tumor growth at the highest dosage of 37.5 mg/kg/day. However, this inhibition did not reach statistical significance, probably due to the high variability in the size of the xenografts and the relatively small number of animals in each group. PC9-GR4 xenografts were found to be less sensitive to S49076, with a reduction of only 21% in tumor growth at 37.5 mg/kg/day (Fig. 3f and Supplementary Fig. 9).

At sacrifice, subcutaneous tumors from PC9-ER were included in paraffin. IHC revealed that the total levels of pH3 in the tumors from S49076-treated mice decreased in a dose-dependent way, while the levels of Ki67 did not. Similarly to cultured PC9-ER, the pH3 staining of xenografts revealed intense staining in mitotic cells and moderate or weak staining in a number of apparently resting cells. The inhibitory effect of S49076 on pH3 levels was found to be circumscribed to the latter (Figs. 3g–h).

**AURKB inhibition induces cell death or senescence.** Inhibition of AURKB has been described to trigger cell cycle arrest and polyploidy. Flow cytometry revealed that barasertib and S49076 at concentrations corresponding to the IC50s induced a significant increase in the fraction of G2/M cells after 24 h in the p.T790M-positive PC9-GR4 and the p.T790M-negative PC9-ER. This effect was accompanied by the appearance of a significant fraction of polyploid cells, particularly in the case of PC9-GR4 (Fig. 4a).

Proliferation assays had revealed that barasertib and S49076 had a strong antiproliferative activity in non-p.T790M-resistant cells. Consequently, we investigated if these drugs could induce cell death following cycle arrest. Flow cytometry experiments after Annexin V and PI staining showed that AURKB inhibition triggered apoptosis/necrosis after 24 h in all the p.T790M-negative cell lines tested, namely PC9-ER, PC9-GR3, and PC9-GR1AZD3. In contrast, no cell death was observed in the p.T790M-positive cells PC9-GR4, PC9-GR1, PC9-GR4AZD1, and H1975 (Figs. 4b–f). Western blotting for cleaved PARP detection confirmed this result (Fig. 4c). In the case of the p.T790M-positive cells, microscopic observation revealed that polyploid cells could survive several days in presence of the drugs.

Following polyploidy, AURKB inhibitors induce senescence in several cell types, via the ATM/Chk2 DNA damage response (DDR). Senescent cells are characterized by permanent growth

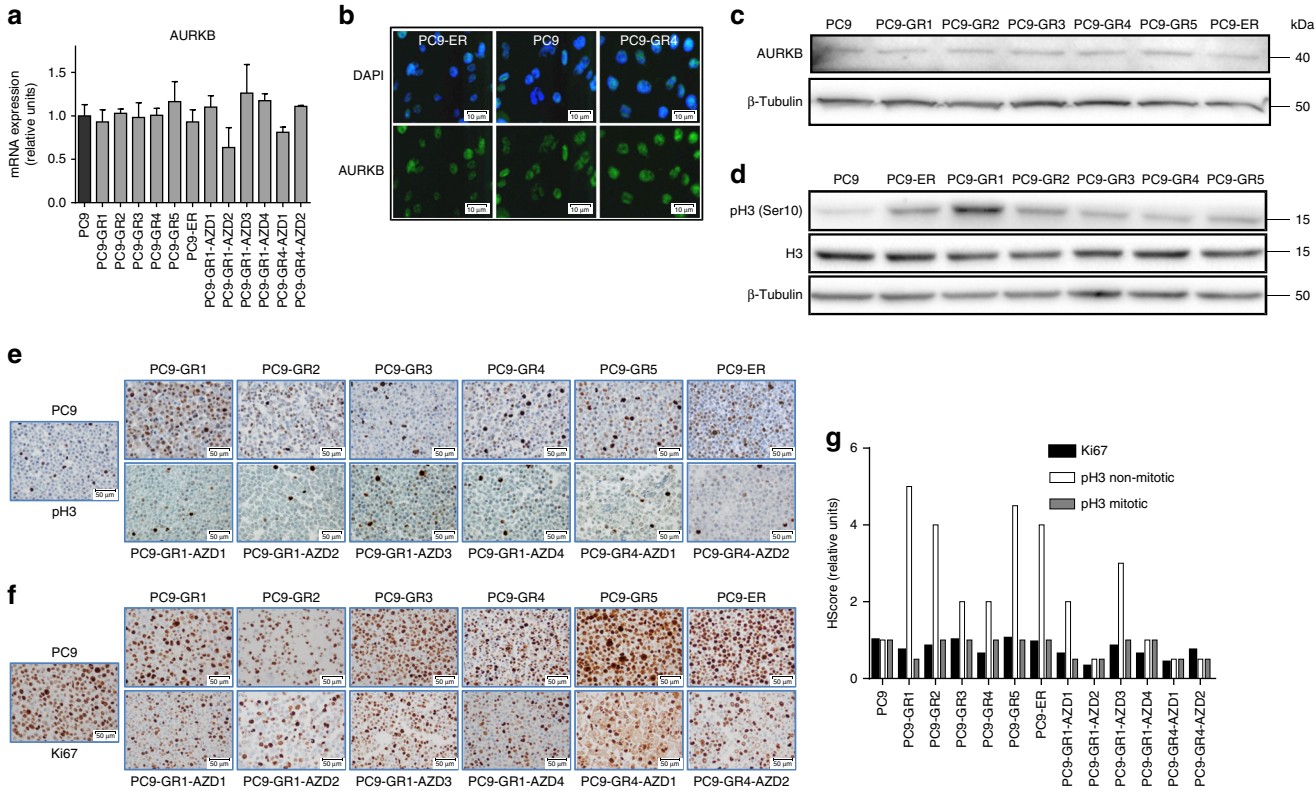

**Fig. 2** AURKB and pH3 in PC9-derived cell lines resistant to EGFR TKIs. **a** Levels of *AURKB* mRNA in the cells of the panel. Values shown are means ± SD of three independent determinations (*n* = 3). **b** Levels of AURKB by ICC in the parental PC9 and the resistant lines PC9-ER (p.T790M negative) and PC9-GR4 (p.T790M positive). Scale bars indicate 10 μm. **c** Levels of AURKB in the parental PC9 and the cell lines resistant to first generation EGFR TKIs by western blotting. **d** Levels of histone H3 and pH3 in the same cell lines by western blotting. **e–g** IHC of pH3 (**e**), Ki67 (**f**) and H-score of Ki67, mitotic pH3 and non-mitotic pH3 (**g**) in the 12 cell lines resistant to EGFR TKIs. Scale bars indicate 50 μm. Values shown are means of two blind determinations performed independently by two expert pathologists (*n* = 2)

arrest, enlarged nucleus and cellular size, vacuolization, over-expression of acidic beta-galactosidase and the senescence-associated secretory phenotype (SASP), mediated by NF-κB and characterized by the secretion of growth factors (i.e., HGF) and other proteins[29]. We used several techniques to determine if AURKB inhibition induced senescence in our resistant cell lines. Beta-galactosidase staining revealed a significant number of positive cells after 48 h treatment with barasertib in all the p.T790M-positive cell lines tested, that included PC9-GR1, PC9-GR4, PC9-GR4AZD1, and H1975 (Figs. 4d–g). Remarkably, cells over-expressing acidic beta-galactosidase were almost invariably polyploid under microscopic examination. Regarding the p.T790M-negative cells, no senescence was apparent in the PC9-GR3 while a weak positivity associated with polyploidy in the few cells surviving at the highest concentration of drug (0.1 μM) was observed in the case of PC9-ER and PC9-GR1-AZD3. Similar results were obtained for S49076 in the PC9-GR4 cells, while no beta-galactosidase positivity was apparent in the PC9-ER (Fig. 4d). Next, we used C12FDG followed by flow cytometry to quantify the effect, and we found a dose-dependent increase in the percentage of beta-galactosidase-positive cells in PC9-GR4 cells, which reached 65% at 1 μM S49076. In contrast, no positivity was observed in PC9-ER (Fig. 4e). Finally, mRNA analyses showed up-regulation of the DDR genes *ATM*, *CHK-2*, and *53BP1*, together with *NF-κB*, in the PC9-GR4 but not in the PC9-ER cells treated with barasertib or S49076 (Fig. 4h). Taken together, these findings confirmed the induction of DDR-associated senescence and NF-κB-mediated SASP in the p.T790M-positive resistant cells.

Senescence is a heterogeneous phenotype, and several examples have been described of pseudo-senescent cells where growth

arrest was reversible[29]. In order to determine if this was also the case in the PC-GR4 cells, barasertib (0.01–0.2 μM) or S49076 (0.1–1 μM) were removed after 2, 3 and 6 days. Growth arrest and morphological changes were found to be permanent after 2 days in cells treated with 0.2 μM barasertib and 1 μM S49076, while lower concentrations required incubation times up to 6 days.

**11–18-derived resistant cells are sensitive to AURKB inhibition.** In order to validate our findings in an additional model of resistance to EGFR TKIs, we used six p.T790M-negative gefitinib-resistant lines (GR1–GR6), derived from 11–18 lung adenocarcinoma cells, which had been previously established in our laboratory (Table 3)[25]. While PC9 carries a deletion in exon 19 of EGFR, the 11–18 cells harbor the *EGFR* p.L858R.

Similarly to the PC9-derived models, four of the 11–18 resistant cell lines showed a significant increase of weak or moderate nuclear pH3 staining (1+ or 2+) in apparently resting cells. In contrast, the H-scores for Ki67 and strong (3+) pH3 staining in mitotic cells were not altered (Figs. 5a–c). Also, while parental 11–18 cells showed an IC50 for barasertib >50 μM in MTT assays, the AURKB inhibitor demonstrated a strong antiproliferative activity in 11–18-GR3 and GR5, with IC50s in the nM range (Fig. 5d, Table 3). 11–18-GR1, GR2, and GR6 cells showed intermediate sensitivity (IC50 10–20 μM), and only 11–18-GR4 cells were completely resistant to the inhibitor. In order to gain insight into the mechanisms underlying these differences, cells were submitted to NGS, revealing the emergence of p.Q61 *NRAS* mutations in 11–18 GR1, GR2, GR4, and GR6 at allelic fractions of 4–22%. In contrast, the two cell lines sensitive

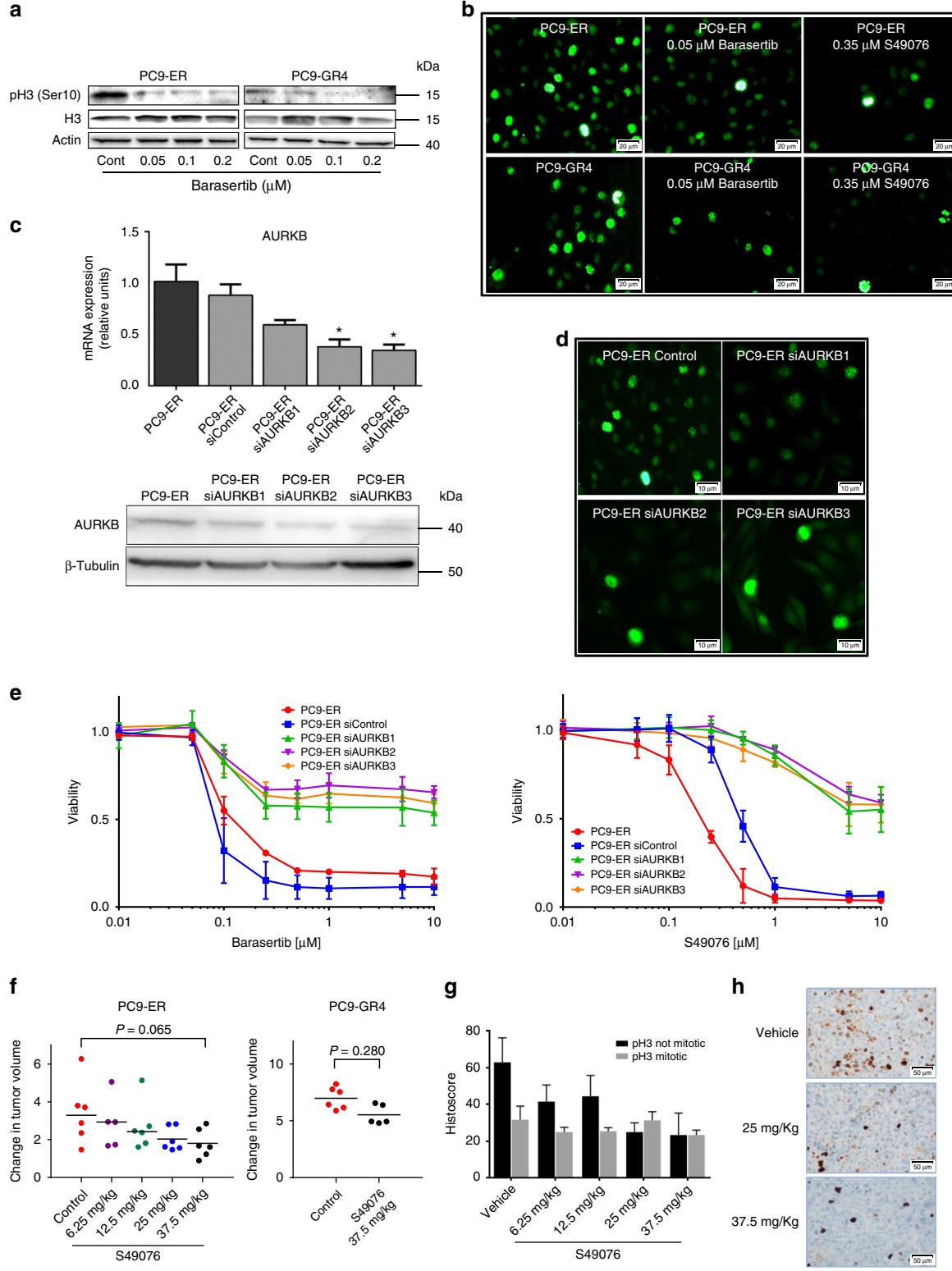

to barasertib (GR3 and GR5) did not show *NRAS* or other acquired mutations (Table 3). Remarkably, the *NRAS* gene showed a 4–5 fold amplification in the entire panel of 11–18 cells, independently of the presence of mutations.

The 11–18-GR2 (*NRAS* p.Q61L) and GR5 cells (wt for *NRAS*) were selected for further work. Similarly to PC9-derived lines, cells treated with barasertib appeared enlarged, vacuolized and multinucleated and comparison of direct counting vs. viability assays in 11–18-GR2 revealed a significant effect of barasertib on

cell numbers, which was not reflected in MTT (Supplementary Fig. 10). Also, cell cycle analyses showed polyploidy followed by cell death exclusively 11–18-GR5, while β–galactosidase staining demonstrated extensive senescence only in 11–18-GR2 (Figs. 5e–f).

**pH3 increases in *EGFR*-mut NSCLC after progression to EGFR TKIs**. We used IHC to determine pH3 levels in 68 pretreatment

**Fig. 3** Inhibition and silencing of *AURKB* in cells and xenografts resistant to EGFR TKIs. **a** Western blot showing the dose-dependent decrease of pH3 levels induced by barasertib at 24 h in PC9-ER and PC9-GR4 cells. **b** ICC showing the inhibition of non-mitotic pH3 by barasertib and the MET/AXL/FGFR/ AURKB inhibitor S49076 after 24 h in PC9-ER and PC9-GR4 cells. Scale bars indicate 20 μm. **c** Levels of *AURKB* mRNA (upper panel) and protein (lower panel) in clones with partial silencing of the *AURKB* gene (PC9-ER siAURKB1–3), compared to parental PC9-ER and PC9-ER cells transfected with a control lentivirus (PC9-ER siControl). Results shown are means ± SD of three independent determinations ($n = 3$) and asterisks indicate statistical significance ($p < 0.05$) in a two-sided Student's *t*-test. **d** ICC showing the inhibition of non-mitotic pH3 in clones with partial silencing of the *AURKB* gene. Scale bars indicate 10 μm. **e** Dose-response curves to barasertib (left) and S49076 (right) of the parental PC9-ER and the PC-ER clones with partial silencing of AURKB. Values shown are means ± SD, experiments were conducted in tri ($n = 3$) or quadruplicates ($n = 4$). In each experiment, every concentration of drug was tested in sextuplicates ($n = 6$). **f** Effects of S49076 on the growth of subcutaneous PC9-ER (left panel) and PC9-GR4 (right panel) xenografts. Each point represents an individual tumor. Tumor dimensions were measured three times per week by digital calipers and volumes estimated according to the formula $V = \pi/6 \times L \times W2$, where $L$ is the long axis and $W$ is the short axis of tumor, respectively. Values are expressed as the fold-change in volume of each individual tumor after the 21 day treatment ($n = 6$ tumors per group). Lines indicated medians, and numbers correspond to levels of significance ($p$) in a Mann–Whitman U test. **g** H-scores of mitotic and non-mitotic pH3 in the tumor xenografts. Values shown are means ± SD ($n = 6$ tumors per group). H-scores of every tumor were evaluated independently by two expert pathologists ($n = 2$). **h** pH3 immunostaining of representative tumor xenografts

tumor samples and 24 re-biopsies after relapse to EGFR TKIs from *EGFR*-mut patients (Figs. 6a–b). The H-score corresponding to mitotic (3+) pH3 did not show significant differences between both groups of samples ($p = 0.135$ in a Mann–Whitney $U$ test), with medians of 15 in both cases. The score for Ki67 showed a similar behavior. In contrast, the median H-score of non-mitotic pH3 was 48 for baseline samples vs. 75 for samples after progression ($p = 0.021$ in a Mann–Whitney $U$ test). Remarkably, only 7% of baseline biopsies had H-scores >100 for non-mitotic pH3 vs. 25% of re-biopsies ($p = 0.023$ in a Z score test).

**High pretreatment pH3 correlates with poor outcome**. Finally, we investigated if pH3 was associated with patient outcome. Of the 68 patients with pretreatment samples mentioned above, 59 had a clinical data available and were used for the analysis (Supplementary Table 2). We found that a high total pH3 H-score associated with poor response to EGFR TKIs and shorter overall survival. Of the 15 patients in the upper quartile, 9 (60%) had progressive or stable disease; in contrast with 16/44 (36%) for the rest of the cohort. Regarding overall survival, it was 15.7 months for the 15 patients with high pH3 vs. 37.6 months for the 44 remaining patients ($p = 0.001$ in a long-rank test; HR = 3.15, CI 95% = 1.98–13.11) (Fig. 6c). When the same analysis was performed for Ki67, mitotic pH3 or non-mitotic pH3 separately, the differences in OS between the groups did not reach statistical significance (Supplementary Table 3).

## Discussion

In this study, we used three panels of cell lines with acquired resistance EGFR TKIs; two were derived from PC9 cells, which carry a deletion in exon 19 of *EGFR*; and one from 11–18 cells, which harbor a p.L858R mutation. The eighteen cell lines of our panels presented a variety of molecular alterations, including acquisition or loss of the p.T790M, emergence of *NRAS* mutations, MET activation and AXL, EGFR, and FGFR1 upregulation. Previous reports have shown that *EGFR*-mut cell lines and tumors with acquired resistance to EGFR TKIs associated with AXL, MET, or FGFR1 are sensitive to inhibitors targeting those proteins[11,30–32]. In contrast, our PC9-derived cell lines did not show sensitivity in vitro to AXL, MET, and FGFR1 inhibitors and *AXL* silencing failed to abolish resistance to EGFR TKIs. However, GAS6 expression was widespread in our resistant cells, suggesting that AXL might play a role in the acquisition of resistance to EGFR TKIs that our relatively simple cell culture models failed to apprehend. In this respect, it has been demonstrated that dependency on AXL in certain cell line models can be easier to recapitulate in xenografts or 3D cultures[33,34]. Regarding

MET, the cells in our panels with dysregulation did not show gene amplification or protein overexpression, only increased phosphorylation of the receptor under basal conditions; a fact that might explain their lack of response to MET inhibitors.

The resistant cell lines not carrying acquired mutations were strongly sensitive to barasertib, a mono-targeted AURKB inhibitor. Tozasertib, an AURKA+ AURKB inhibitor, and S49076, an inhibitor targeting AXL, MET, FGFR, and AURKB in Phase I/II clinical trials[27,35], showed a similar behavior. Active AURKB phosphorylates histone H3, its main substrate, on Ser10. Despite the diversity of molecular alterations present in our resistant cell lines, we observed a widespread increase of pH3 levels not associated with a higher proliferation rate and circumscribed to the nuclei of apparently resting cells, where pH3 often appeared in foci. Such pH3 foci, often around nucleoli, have also been described in other cell types[36]. Partial silencing or drug inhibition of AURKB significantly decreased pH3 levels in apparently resting cells, indicating that activation of AURKB was the cause of the increase of pH3 in our panels of resistant cells. Amplification or overexpression of AURKB has been described in some solid tumors, where it has been correlated with poor prognosis[18,20]. However, gene expression and Western blotting experiments demonstrated that the activation of AURKB in our cell models of resistance was not associated with an upregulation at the mRNA or protein level. Also, the partial silencing of the *AURKB* gene did not re-sensitize cells to EGFR TKIs, suggesting that AURKB activation and increased pH3 levels were not responsible for the acquisition of resistance, but rather associated with it.

The mechanism responsible for increased AURKB activity in apparently non-mitotic cells remains to be elucidated. At early G2, AURKB is activated and extensively phosphorylates H3. After mitosis, AURKB undergoes targeted proteolysis by the anaphase-promoting complex/cyclosome (APC/C) ubiquitin ligase[37] and, during interphase, AURKB activity and pH3 levels remain low through mechanisms not fully understood[38]. At this respect, the phosphatases PP1 and PP2A can inactivate AURKB[39], and AURKB activity and/or H3 Ser10 phosphorylation has been proposed to be actively suppressed by PP1 during interphase[36]. The observation that total AURKB levels in the parental and resistant cells are undistinguishable suggests that the increased AURKB activity and pH3 in the latter are a consequence of a deregulated dephosphorylation during interphase rather than impaired ubiquitination after mitosis.

Inhibition or silencing of AURKB has been demonstrated to induce G2/M arrest and polyploidy in several cell line models[28,40]. Western blotting and flow-cytometry experiments showed that this was also the case in our EGFR-TKI-resistant cell lines, irrespective of the p.T790M or *NRAS* status. However, treatment with barasertib or S49076 had strong antitumor

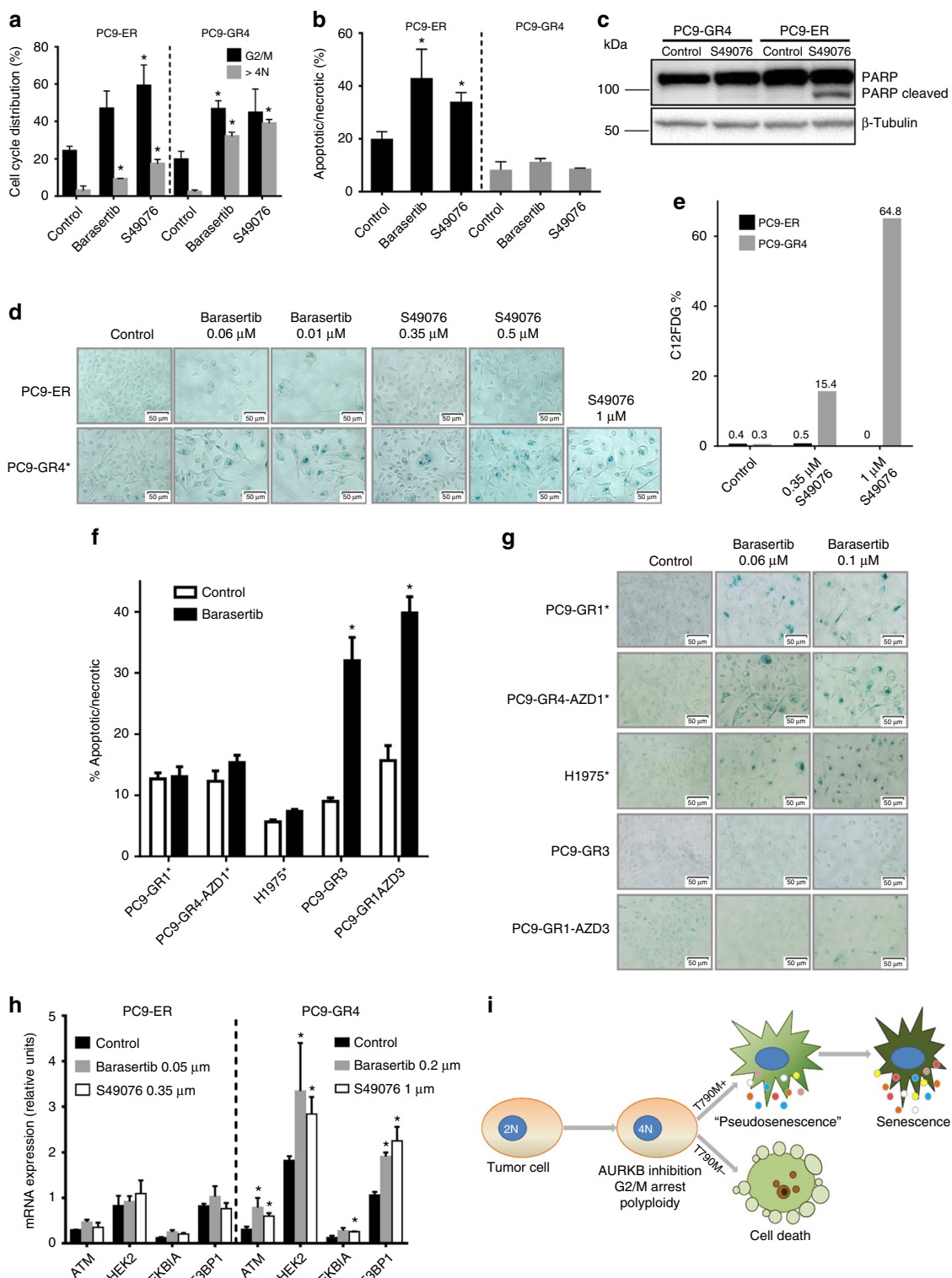

effects only in absence of p.T790M or *NRAS* mutations. Annexin V analyses revealed that both drugs induced apoptosis exclusively in p.T790M and *NRAS*-negative resistant cells. In positive cells, AURKB inhibition and polyploidy were followed by profound changes characteristic of the senescent phenotype, which were clearly observable after 2 days and included growth arrest, enlarged cellular size, cell flattening, irregular nuclei and overexpression of acidic β-galactosidase. However, if barasertib at <0.2 μM or S49076 at <1 μM were

removed after less than 6 days, the p.T790M-positive PC9-GR4 resumed growth, indicating that AURKB inhibition initially induced a pseudo-senescent state where growth arrest was not yet irreversible[29]. In melanoma cells, AURKB has been shown to induce senescence through the DDR and the transcription factor NF-κB[41]. The up-regulation of *NF-κB* and the DDR genes *ATM*, *CHK–2*, and *53BP1* observed after treatment with barasertib and S49076 indicates that this is also the case in PC9-GR4.

**Fig. 4** Inhibition and silencing of *AURKB* in cell lines resistant to EGFR TKIs (cont). **a** Results of flow-cytometry experiments in PC9-ER (p.T790M negative) and PC9-GR4 (p.T790M positive) showing the induction of G2/M arrest and polyploidy after 24 h treatment with barasertib (0.06 µM) or S49076 (0.35 µM). Values shown are means ± SD of three independent determinations (n = 3) and asterisks indicate statistical significance (p < 0.05) in a two-sided Student's t-test. **b** Results of flow-cytometry experiments showing the induction of cell death in PC9-ER but not in PC9-GR4 after 24 h treatment with barasertib (0.06 µM) or S49076 (0.35 µM). Values shown are means ± SD of three independent determinations (n = 3) and asterisks indicate statistical significance (p < 0.05) in a two-sided Student's t-test. **c** Western blot analysis of poly [ADP-ribose] polymerase 1 (PARP) cleavage in PC9-ER and PC9-GR4 after 24 h with S49076 (0.35 µM). **d** Acidic beta-galactosidase staining of PC9-ER and PC9-GR4 cells after 48 h with barasertib or S49076, showing the induction of senescence-associated activity in PC9-GR4. Scale bars indicate 50 µm. **e** Percentage of cells with active acidic beta-galactosidase, as measured by C12FDG fluorescent staining followed by flow cytometry, in PC9-ER and PC9-GR4 cells treated with S49076 for 48 h. **f** Results of flow-cytometry experiments of PC9-GR1, PC9-GR4-AZD1, H1975 (p.T790M positive), PC9-GR3, and PC9-GR1-AZD3 (p.T790M negative), showing the induction of apoptosis only in p.T790M-negative cells after 24 h treatment with barasertib (0.06 µM). Values shown are means ± SD of three independent determinations (n = 3) and asterisks indicate statistical significance (p < 0.05) in a two-sided Student's t-test. **g** Acidic beta-galactosidase staining of the same cells after 48 h with barasertib, showing the induction of senescence-associated activity exclusively in p.T790M-positive cells. Scale bars indicate 10 µm. **h** Changes in the mRNA levels of selected genes after a 6-day treatment with S49076 or barasertib. Values shown are means ± SD of three independent determinations (n = 3) and asterisks indicate statistical significance (p < 0.05) in a two-sided Student's t-test. **i** Proposed model to explain the effects of S49076 and barasertib in EGFR-TKI-resistant cells

---

**Table 3 Characteristics (left) and sensitivity (right) to gefitinib and barasertib of the 11–18 derived cell lines used in the study**

| Cell line | Doubling time (h) | EGFR L858R | EGFR T790M | EGFR overexpression | AXL overexpression | MET activation | FGFR1 overexpression | NRAS amplification (fold) | NRAS mutations Q61L/R/K | Gefitinib (EGFR) IC50 (µM) | Barasertib (AURKB) IC50 (µM) |
|---|---|---|---|---|---|---|---|---|---|---|---|
| 11–18 | 23 ± 4 | + | − | − | − | − | − | 4 | − | 0.2 | >50 |
| 11–18 GR1 | 27 ± 4 | + | − | + | − | − | − | 5 | 22% (Q61R) | 27.1 | 13 |
| 11–18 GR2 | 25 ± 2 | + | − | + | − | + | − | 4 | 9% (Q61L) | 8.0 | >50 |
| 11–18 GR3 | 24 ± 2 | + | − | + | − | − | − | 4 | − | 27.4 | 0.06 |
| 11–18 GR4 | 22 ± 1 | + | − | + | − | + | ++ | 5 | 4% (Q61L) | 20.6 | 14 |
| 11–18 GR5 | 27 ± 4 | + | − | + | − | − | − | 3 | − | 20.8 | 0.06 |
| 11–18 GR6 | 22 ± 1 | + | − | + | ++ | − | − | 4 | 9% (Q61K) | 8.1 | 18 |

The targets of the drugs are indicated between parentheses. Doubling times are expressed as means ± SD of at least three different determinations

---

After growth arrest and polyploidy, AURKB inhibition triggers cell death in lung cancer, leukemia, prostate cancer, or neuroblastoma cell[40,42,43] but senescence in fibroblasts, endothelial or melanoma cells[41,44]. In the case of fibroblasts, senescence is the result of cell cycle blockade by AURKB inhibition in presence of persistent strong mitogenic stimuli[45]. Similarly, it has been shown that aberrant replication or DNA damage, followed by growth arrest, leads to senescence in cells with oncogenes activated by mutations or other mechanisms[46–49]. In the context of resistance to TKIs, an AURKA/AURKB inhibitor has been reported to induce senescence in chronic myeloid leukemia cells carrying the imatinib-resistant T315I mutation in *ABL*[50]. This could also be the case in our EGFR-TKI-resistant cells harboring acquired p.T790M or *NRAS* mutations. These mutations induce persistent and strong mitogenic stimuli that, in presence of AURKB inhibition and cell cycle blockade, could trigger first a pseudo-senescent and latter a truly senescent state. Senescent cells can survive for long periods of time, explaining the limited antitumor effects of S49076 or barasertib in p.T790M- or *NRAS*-positive cells. Conversely, in resistant cells without acquired mutations, cell cycle arrest in absence of oncogenic activation would induce cell death (Fig. 4i). Further support of this hypothesis comes from the observation that, in most p.T790M-negative lines of our PC9-derived panels, a small fraction of cells (10–30% in MTT assays) survives to micromolar concentrations of barasertib but not of S49076, showing a senescent phenotype (Table 2, Fig. 4d). We can speculate that, in resistant cells without acquired mutations, AXL, FGFR1, and EGFR overexpression or MET activation provide weak oncogenic stimuli. In presence of barasertib, which inhibits exclusively AURKB, these weak signals can direct a minor fraction of cells to senescence and survival. In contrast,

being an MET/AXL/FGFR1/AURKB inhibitor, S49076 can suppress these stimuli and lead all cells to death.

The increase in pH3 associated with resistance to EGFR TKIs was not circumscribed to cell line models. In tumor samples from *EGFR*-mut NSCLC patients, the levels of non-mitotic pH3 were significantly elevated after progression to TKIs. In particular, 25% of tumors at progression had an H-score ≥ 100 vs. 7% at presentation. Similarly to resistant cell lines, increased non-mitotic pH3 coexisted with the p.T790M mutation, AXL overexpression or MET amplification. Finally, patients expressing high pretreatment levels of pH3 had a worse response rate to EGFR TKIs and a significantly shorter overall survival. These results were obtained in a relatively small number of samples. If confirmed in a larger cohort, they would indicate that AURKB activation and pH3 levels are involved in both intrinsic and acquired resistance to EGFR TKIs.

In summary, we have demonstrated that acquisition of resistance to EGFR TKIs in *EGFR*-mut cell line models and NSCLC patients is often associated with AURKB activation and increased levels of histone H3 phosphorylation. Inhibition of AURKB in these cell models leads to cell cycle arrest and polyploidy, followed by extensive cell death when the p.T790M or other acquired mutations are absent, or by senescence if they are present. Taken together, our results indicate that AURKB is a potential therapeutic target in NSCLC patients progressing on EGFR TKIs and not harboring resistance mutations.

## Methods

**Cell culture, proliferation assays, and cell cycle analyses**. The H1975 cell line was purchased from the ATCC. Parental PC9 cells were provided by F. Hoffman-La Roche Ltd (Basel, Switzerland) with the authorization of Dr. Mayumi Ono

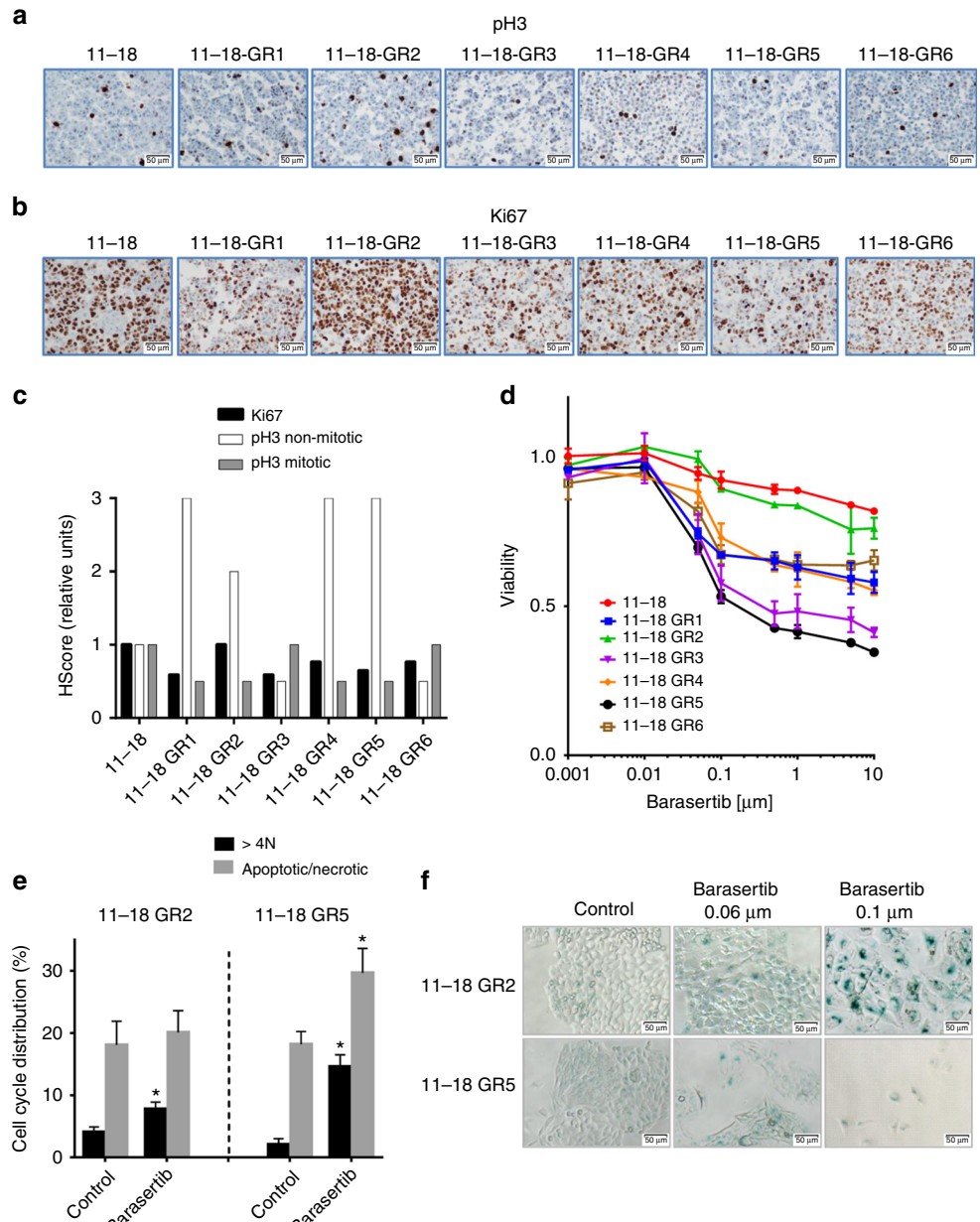

**Fig. 5** Levels of pH3 and *AURKB* inhibition in 11–18 derived cell lines resistant to gefitinib. **a–c** IHC of pH3 (**a**), Ki67 (**b**), and H-score (**c**) of Ki67, mitotic pH3 and non-mitotic pH3 in the six *EGFR*-mut cell lines resistant to EGFR TKIs. Scale bars indicate 50 μm. Values shown are means of two blind determinations performed independently by two expert pathologists ($n = 2$). **d** Dose-response curves of the parental and the EGFR-TKI-resistant cells to the AURKB inhibitor barasertib. Values shown are means ± SD, experiments were conducted in tripicates ($n = 3$). In each experiment, every concentration of drug was tested in sextuplicates ($n = 6$). **e** Results of flow-cytometry experiments in 11–18-GR2 (*NRAS* Q61L) and GR5 (*NRAS* wt) showing polyploidy after 24 h treatment with barasertib (0.1 μM) and apoptosis in 11–18-GR5 but not in GR2. Results shown are means ± SD of three independent determinations ($n = 3$) and asterisks indicate statistical significance ($p < 0.05$) in a two-sided Student's *t*-test. **f** Acidic beta-galactosidase staining of 11–18-GR2 and GR5 cells after 48 h with barasertib, showing the induction of senescence-associated activity in 11–18-GR2. Scale bars indicate 50 μm

(Kyushu University, Fukuoka, Japan). Parental 11–18 cells were kindly provided by Dr. Mayumi Ono. Resistant cells were derived by culturing parental cells with EGFR TKIs, starting at the IC50. Concentrations were increased in a stepwise way when proliferation resumed and cells could be trypsinized and re-plated. Fresh drug was added every 72–96 h. Resistant cells were maintained as polyclonal populations under concentrations of drug ≥2.5 μM. Cell lines were authenticated by analyzing >20 polymorphisms by NGS (see below). In all cases, the genotypes and allelic fractions of parental and resistant cells were identical. S49076 was kindly provided by Servier (Suresnes, France). The rest of inhibitors were purchased from Selleck Chemicals (Houston, TX, USA) or MedChem Express (Monmouth Junction, NJ, USA).

Tumor cells were grown in a humidified atmosphere with 5% $CO_2$ at 37 °C in RPMI1640 + 10% fetal bovine serum (FBS), 50 μg/mL penicillin-streptomycin, and

2 mM L-Glutamine and routinely tested for mycoplasm contamination. For proliferation assays, cells were seeded at a density of 2000 (PC9) or 4000 (resistant) cells per well in 96-well plates, allowed to attach for 24 h and treated with drugs for 72 h unless otherwise indicated. For calculation of doubling times, cells were grown without drugs for 0, 24, 48, and 72 h following attachment. After treatment, cells were incubated with medium containing 0.75 mg/mL of 3-(4,5-dimethylthiazol-2-yl)−2,5-diphenyltetrazolium bromide (MTT) for 1–2 h at 37 °C, medium was removed and formazan crystals dissolved in 100 μL DMSO. Cell numbers were estimated by measuring the absorbance at 495 nm, using an Anthos 2020 microplate reader (Biochrom Ltd, Cambridge, UK). Data were derived from at least three independent experiments. In some experiments, direct counting in a hemocytometer after trypan blue staining was additionally used. For cell cycle analyses, cells were plated in T-25 flasks, allowed to attach, treated with drugs in

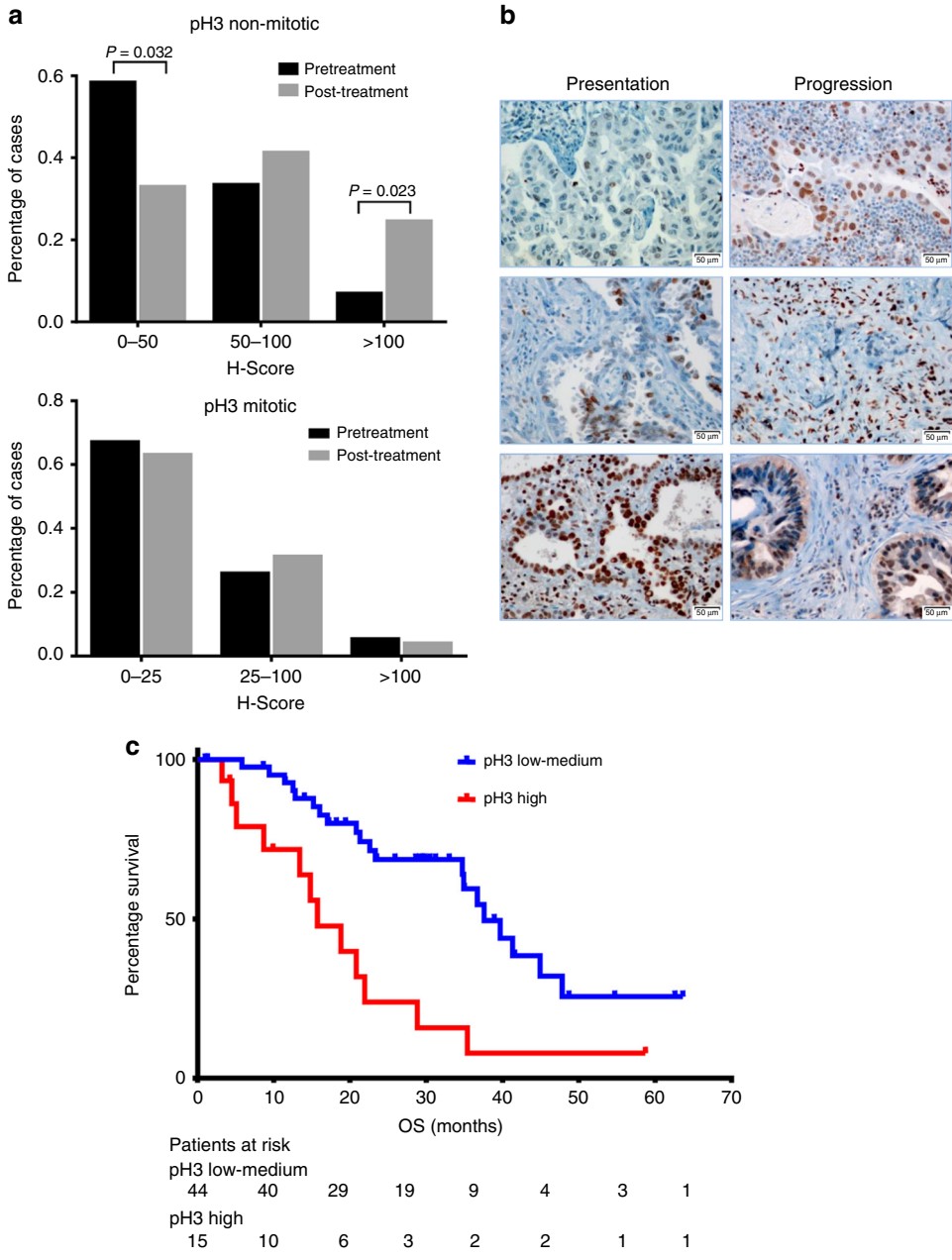

**Fig. 6** pH3 in *EGFR*-mut NSCLC patients. **a** H-scores of non-mitotic (upper panel) and mitotic pH3 (lower panel) and in samples from EGFR-mut NSCLC patients baseline ($n = 68$, black bars) and after progression to EGFR TKIs ($n = 24$, gray bars). H-scores of every tumor were evaluated independently by two expert pathologists ($n = 2$). Numbers indicate levels of significance ($p$) in a Z score test. **b** IHC staining of pH3 of representative tumors at presentation and after progression to EGFR TKIs. Scale bars indicate 50 μm. **c** OS according to overall pH3 H-score at presentation in 59 *EGFR*-mut NSCLC patients treated with first-line EGFR TKIs. Median OS was 15.7 (CI 95% = 8.9–22.5) months for the 15 patients with high pH3 (Q4, red line) vs. 37.6 months (CI 95% = 31.6–43.5) for the 44 patients with moderate or low pH3 (Q1–Q3, blue line) ($p = 0.001$ in a two-sided log-rank test; HR = 3.15, CI 95% = 1.98–13.11)

RPMI + 10% FBS, trypsinized, and centrifuged. Cell pellets were resuspended in PBS, fixed in 70% ethanol and incubated o/n at −20 °C. Fixed cells were subsequently centrifuged at $400 \times g$ for 5 min at 4 °C, resuspended in 250 μL of PBS with 50 μg/mL RNAse A (Sigma–Aldrich), incubated for 1 h at 37 °C, and stained with propidium iodide (PI) (Roche Diagnostics) for 30 min at room temperature.

**Western blot, immunocytochemistry, and immunohistochemistry**. For Western blotting, cells were seeded in T-75 flasks, allowed to attach o/n and treated for 2–24 h in RPMI + 10% FBS, unless otherwise indicated. Then, cultures were washed twice with cold PBS, scrapped into RIPA buffer (Cell Signaling Technology, Danvers, MO, USA) (20 mM Tris-HCl pH 7.5, 150 mM NaCl, 1 mM Na$_2$EDTA, 1 mM EGTA, 1% NP-40, 1% sodium deoxycholate, 2.5 mM sodium pyrophosphate, 1 mM β-glycerophosphate, 1 mM Na$_3$VO$_4$, 1 μg/ml leupeptin, 2 mM PMSF) with a protease inhibitors cocktail (Roche Diagnostics, Basel, Switzerland) and passed through an insulin syringe. Lysates were transferred to a microfuge tube, incubated

on ice 15 min, centrifuged for 10 min at $18,500 \times g$ and immediately analyzed or frozen at –80 °C. Protein extracts (25 μg) were boiled in Laemmly buffer (NuPAGE-LDS sample buffer 4×; Invitrogen, Carlsbad, CA), resolved in SDS-polyacrylamide gels and transferred to a PVDF membrane (Bio-Rad, Hercules, CA). Membranes were incubated for 1 h in Odyssey blocking buffer (Li-Cor Biosciences, Lincoln, NE) or Phosphoblocker reagent (Cell Biolabs Inc, San Diego, CA) for phosphorylated proteins. After blocking, membranes were cut, incubated with primary antibodies (Supplementary Table 4) o/n at 4 °C, washed three times for 5 min each in PBS-Tween 0.1%, and incubated for 2 h with secondary antibody (anti-rabbit or anti-mouse IgG horseradish peroxidase-conjugated secondary antibody; GE Healthcare, New York, NY). Finally, after three additional washes, membranes were incubated with Supersignal Chemiluminiscence substrate (Thermofisher Scientific, Waltham, MA, USA) and read with a a Bio-Rad ChemiDocMP Imaging System.

Fluorescent Immunocytochemistry (ICC) was performed on cells seeded in 24 well plates ($2.5 \times 10^4$ per well) with round cover slips and allowed to attach

overnight. Cover slips were then removed, washed with PBS, fixed with 4% paraformaldehyde for 15 min and treated with cold methanol. After a wash in ICC buffer (PBS 1×, 0.1% TX-100, 0.05% Sodium azide, 0.5% BSA), they were blocked in ICC + 10% FBS for 20 min at room temperature and incubated overnight at 4 °C with primary antibodies against the indicated proteins. Next, after three wash steps, Alexa Fluor 488-coupled secondary antibodies (Thermo Fisher Scientific) were added and incubated at room temperature for 2 h. Finally, cover slips were washed three times with ICC, counterstained with DAPI, washed with water and mounted on glass slides with Vectashield (Vector Laboratories, Burlingame, CA, USA).

Immunohistochemistry (IHC) was performed on 5-μm sections of formalin-fixed, paraffin embedded (FFPE) blocks using an automated immunostainer (BenchMark ULTRA, Ventana Medical Systems, Oro Valley, AZ, USA). Protein expression was quantified in a blind manner by two independent pathologists, using the histoscore (H-score) method, which evaluates the intensity of staining (0, non-staining; 1, weak; 2, median; or 3, strong) and the percentage of positive cells. The antibodies used in this study are presented in the Supplementary Table 4. Histological images were obtained using a ×20 objective, unless otherwise indicated.

**Cell death and senescence assays**. For cell death analyses, the Annexin-V-FLUOS (An) staining kit (Roche Diagnostics) was employed, according to the manufacturer's instructions. Stained cells were analyzed with a FACSCanto II cytometer (BD Biosciences, Franklin Lakes, NJ) using the FACSDiva software version 6.1.2. The combination of Annexin V and PI was used to differentiate four cell populations; namely, viable cells (An−/PI−), early apoptotic (An+/PI−), necrotic (An−/PI+), and later apoptotic/necrotic (An+/PI+). For the initial senescence studies, cells were treated for 2 h to 8 days in 6-well plates, and the Senescence Beta-galactosidase Staining kit (Cell Signaling) was be used to evaluate the presence of senescent cells by direct observation under the microscope. In subsequent studies, the percentage of senescent cells was quantified by flow cytometry. Cells ($3 \times 10^4$) were grown in T-25 flasks and treated with selected drugs and incubated for 1 h with fresh RPMI + 10%FBS with 100 nM bafilomycin A1 to neutralize the acidic pH of lysosomes. Finally, 5-dodecanoylaminofluorescein di-β-D-galactopyranoside (C12FDG) was added at a final concentration of 33 μM for 2 h and cells were submitted to flow cytometry using standard procedures. Gating strategies for flow-cytometry analyses of cell cycle, death and senescence are presented in Supplementary Fig. 11.

**NGS, mRNA analysis, and gene silencing**. Purified DNA (16.5 μL) was used as a template for NGS analysis using the Gene Reader platform and GeneRead® QIAact Lung DNA Panel (QIAGEN, Hilden, Germany), according to manufacturer's instructions. The panel targets 549 variant positions in 17 selected genes frequently altered in lung tumors (*AKT1, ALK, BRAF, DDR2, EGFR, ERBB2/HER2, ESR1, KIT, KRAS, MAP2K1, MET, NRAS, NTRK1, PDGFRA, PIK3CA, PTEN, ROS1*) and detects copy number variations (CNV) in five genes (*EGFR, FGFR1, ERBB2/HER2, MET, RICTOR*).

RNA was isolated by standard procedures[25]. Primer and probe sets (Supplementary Table 5) were designed using Primer Express 3.0 Software (Applied Biosystems) according to their reference sequences (http://www.ncbi.nlm.nih.gov/LocusLink). Quantification of gene expression was performed using the ABI Prism 7900HT Sequence Detection System (Applied Biosystems). Expression levels were calculated using to the comparative ΔΔCt method[25]. Commercial RNA controls were used as calibrators (Liver and Lung; Agilent Technologies, Santa Clara, CA, USA). For each cell line, a minimum of three independent experiments were performed.

Silencing of *AXL* and *AURKB* was achieved by stable transfection using shRNA Lentiviral Transduction Particles (Sigma–Aldrich) or with pLV CRISPR-based lentivirus encoding Cas9, a puromycin-resistance gene and a sgRNA targeting *AURKB*. Non-target shRNA or sgRNA particles were used as controls. Transfections were performed on 96-well plates by adding the particles (MOI 0.5–5) to exponentially growing, 50–70% confluent cells in presence of hexadimethrine bromide (8 μg/mL) (Sigma–Aldrich). After a 24 h incubation period, the viral particle-containing medium was replaced by fresh RPMI + 10% FBS. After an additional 24 h, new medium was added, containing the appropriate concentration of puromycin (Sigma–Aldrich) for selection of transduced cells (1 μg/mL). Resistant colonies were picked up, expanded in medium with 1 μg/mL puromycin and assayed for the silencing of the target gene by western blotting and/or mRNA expression analyses. Silenced clones were authenticated as described under cell culture.

**Animal studies**. All animal experiments were approved by the Ethical Committee of Animal Experimentation of the Parc Científic de Barcelona (PCB) following the guidance of the Association for Assessment and Accreditation of Laboratory Animal Care (AAALAC, Unit 1155). All relevant ethical regulations for animal testing and research were complied with. Mice were inoculated subcutaneously with $5 \times 10^6$ PC9-ER or GR4 cells in 0.3 mL of PBS with Matrigel. Treatments were started when the average tumor size reached 150–200 mm³ and mice were assigned into groups ($n = 6$) using randomized block design based upon their tumor volumes. S49076 was dissolved in 1% (w/v) Hydroxy Ethyl Cellulose (HEC) in

100 mM acetate buffer pH 4.5 and orally administered twice a day with a maximum dose of 37.5 mg/kg bid. Tumor dimensions were measured blindly three times. At sacrifice, tumors were excised and included in paraffin for IHC analyses. The investigators were blinded for the evaluation of the results.

**Patients**. The patient cohort consisted of *EGFR*-mutated NSCLC patients diagnosed between 2006 and 2017 in the Dexeus Quirón University Hospital (Barcelona, Spain) and Fundación Santa Fe de Bogotá (Colombia). Studies were conducted in accordance with the Declaration of Helsinki and all relevant ethical regulations for work with human participants, under an approved protocol of the Institutional Review Board of Dexeus Quirón University Hospital. Samples were de-identified for patient confidentiality and informed written consent, also approved by the Institutional Review Board of Dexeus Quirón University, was obtained from all subjects.

**Statistical analysis**. The results were analyzed in GraphPad Prism v6.0 using appropriate statistical tests. $p$-values of <0.05 were considered to be statistically significant.

**Reporting summary**. Further information on experimental design is available in the Nature Research Reporting Summary linked to this article.

## Data availability

The NGS data generated in this study have been deposited in the in the Sequence Read Archive (SAR) of the National Center for Biotechnology Information (NCBI), under the accession codes PRJNA524804 (project) and SAMN11035311-SAMN11035324 (individual samples). The source data underlying Figs. 1e, 2a, 2g, 3c, 3e-g, 4a, 4b, 4e, 4f, 4h, 5c-e, 6a, 6c and Supplementary Figs. 1b, 2, 3c, 3d, 4a, 4b, 5b, 5c, 6a, 6c, 7b, 7c, 9a, 9b, 10b, 10c are provided as a source data file. The file contains uncropped and unprocessed scans of the western blots presented in Figs. 1b, 2c, 2d and 3a. The source data file has also been deposited in the Open Science Framework (OSF) repository under the unique identifier DOI 10.17605/OSF.IO/JW4C7. The authors declare that all other data supporting the findings of this study are available within the main article and its Supplementary Information file or from corresponding authors upon reasonable request. A reporting summary for this article is available as Supplementary Information file.

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

## Acknowledgements

Financial support for this study was provided by the Institut de Recherches Internationales Servier (Suresnes, France)

## Author contributions

J.B.A., V.C., M.S., F.C., M.B., Ra.R. and M.A.M.V. conceived and designed the study. J.B.A., J.C.S., A.G.C., S.R., C.T., Ru.R., J.C., S.G.R., C.C.S. and M.A.M.V. conducted the experiments, acquired, and analyzed data. S.V., A.F.C. and N.K. provided clinical samples and collected clinical information. J.B.A., V.C., M.S., J.C.S., S.G.R., Ra.R. and M.A.M.V. wrote, reviewed, and/or revised the manuscript.

## Additional information

**Competing interests:** V.C., M.S., F.C. and M.B. were employees of Servier at the time of this study. The remaining authors declare no competing interests.

