## [Peer Review File · Nature Communications]

Reviewers' comments:

Reviewer #1 (Remarks to the Author):

In this manuscript, Jordi Bertran-Alamillo et al. uncover AURKB as a potential target in acquired resistance to EGFR-TKIs. They established multiple EGFR TKI resistant cell lines based on the two cell lines, PC9 and 11-18. T790M negative resistant cell lines were characterized by variable upregulation of AXL, MET and FGFR. Using multitargeted and more selective inhibitors they narrowed down AURKB inhibition as having the most potent impact on PC9-derived non-T790M cell lines. Next, they identified pH3 levels as being elevated in 8/12 cell lines, independently of proliferation rate, and used this as an on-target marker of effective AURKB inhibition. In vivo treatment of PC9-ER xenografts confirmed dose dependent growth inhibition and reduction in pH3, though results were just below the level of statistical significance. They next show that T790M positive cells undergo senescence in response to AURKB inhibition, while T790M negative cells undergo cell death. Finally, patients tissue samples showed that pH3 score was associated with poor response to EGFR-TKI. Overall this is a nice study that supports AURKB as a new target in EGFR-mutated lung cancers. However, the results would be strengthened by addressing the concerns below.

Major points

1. Figure legends are missing from the main figures, which makes it difficult to fully interpret them
2. The model that T790M positive cells and T790M negative resistant cells differ in their response to AURKB inhibition is based on only a few cell lines. It should be possible to compare Barasertib SA-b-gal vs PARP cleavage across a larger panel of cell lines. How robust really is this relationship? Also, what is the response of H1975 cells that harbor pre-existing L858R and T790M to Barasertib?
3. Similarly, PC-9-GR1 cells are less sensitive to AURKB inhibition compared with the PC9-GR1-AZD (osimertinib-treated) derivative lines. Is this because osimertinib treatment has more fully eradicated T790M clones and there is a greater apoptotic response?
4. The partial knockdown of AURKB creates a mitotic defect, likely slowing the growth of PC9ER cells - the barasertib drug sensitivity results could be confounded by this. What about the impact of over-expressing of AURKB? Or ideally over-expression of barasertib drug resistant allele?

Minor points

1. Supplementary Fig S6A is mislabeled as tumor weight instead of change in tumor volume

Reviewer #2 (Remarks to the Author):

NSCLC tumors with EGFR mutations often relapse to EGFR TKI therapy due to emergence of EGFR-T790M mutation or other resistant gene dysregulation. In this study, the authors showed EGFR TKI resistant cell lines without the T790M or other resistant mutations are more sensitive to AURKB inhibition than these EGFR TKI resistant cell lines with these resistant mutations. Furthermore, through mechanistic studies, they found AURKB inhibition reduces pH3 level and trigger G1/S arrest and polyploidy in resistant cells, and then causes cell death in non-T790M resistant cells, but only leads to senescence in T790M resistant cells. Their results reveal that AURKB activation is associated with acquired resistance in EGFR mutant lung cancer cells, and that AURKB inhibition is a potential therapeutic target in EGFR TKI-resistant NSCLC tumors without T790 M or other resistance mutations. Undoubtedly, these findings are very important in this field, but some conclusions are not strongly supported by their data, and the following aspects of this manuscript need to be addressed to strengthen the manuscript.

Comments:

1. Figure legends should be included into the manuscript.
2. In Fig.1D, GAS6 staining pattern is different in each cell line. For PC9-GR1-AZD2 and PC9-GR4-AZD1, most cells are cytoplasm staining. But for other lines, most cells are nuclear staining. As we know, GAS6 should be cytoplasm staining.
3. In Fig.2B, AURKB expression in PC9-GR4 looks stronger than PC9-ER and PC9. For PC9-GR4, every cell has strong staining, but for PC9-ER and PC9, a lot of cells show much weaker staining. If AURKB is the most important gene in this story, the authors should do western blot to compare AURKB expression in each line. mRNA level showed in Fig.2A can't show us the real AURKB level.
4. The authors compared PC9-ER with PC9-GR4, and also compared 11-18-GR5 with 11-18-GR2. From Fig.2E and Fig.5B, we can know the proliferation rate between PC9-ER and PC9-GR4, as well as the proliferation rate between 11-18-GR5 and 11-18-GR2 are big different. For the IC50 comparison, it's hard to exclude the influence derived from different cell proliferation rate.
5. For Fig.3E, it's hard to believe that PC9-ER with partial AURKB knockdown is resistant to AURKB inhibition. If AURKB is the major effector in the resistance, AURKB partial knockdown plus AURKB inhibitor should also show good effect to inhibit cell viability. As well as, Fig.3C showed AURKB partial knockdown, but it's the expression level of cell pool. Some cells may have very high AURKB knockdown efficiency, and some other cells may have very low AURKB knockdown efficiency, just like the pattern showed in Fig. 3D. If so, cells with high AURKB knockdown efficiency and cells

with low AURKB knockdown efficiency will perform big different. Under this condition, it's better to use CRISPR system and select half knockout clones for the experiment.

6. For Fig. 3F, the authors only showed in vivo S49076 treatment for one cell line. They should compare PC9-ER with PCR-GR4 tumors. If PC9-ER tumors are more sensitive to AURKB inhibition than PC9-GR4, which will help them to support their conclusion.

7. After AURKB inhibition, T790M resistant cells will finally go senescence, and non-T790M resistant cells will go apoptosis. Cells go senescence, they can't proliferate although they may be still alive, so the final tumor inhibition effect by AURKB inhibition may be very similar. The in vitro experiment only lasts for a few days, and we can't see the long term effect. The authors should do more in vivo treatment study to support their conclusion.

8. In Fig.S4D, PC9-ER(direct count) and PC9-GR4(direct count) didn't show different sensitivity to Barasertib. In Fig.S7B, 11-8 GR2 (number of cells) and 11-18 GR5 (number of cells) didn't show different sensitivity to Barasertib. These results are not consistent with the author's description in the manuscript.

9. In Table 2, PC9-GR2 are T790M negative, but it's not sensitive to Barasertib; PC9-GR4-AZD2 only have 0.03% T790M allelic fraction which is very close to negative, but it's also not sensitive to Barasertib. These two cases are not consistent with the author's points. Besides AURKB activation, any other mechanisms involved in this process?

10. For 11-18 derived resistant cells, the emergence of Q61 NRAS mutations in 11-18 GR1, GR2, GR4, GR6 is 4-22% allelic fractions, which means most cells in each line don't have NRAS mutation, so it's hard to understand these very small population mediates the resistance to AURKB inhibition in IC50 assay which only lasts for a few days. The same concern is also exist in PC-9 based experiments.

Reviewer #1

Major points

1- Figure legends are missing from the main figures, which makes it difficult to fully interpret them

Our reply

Figure legends have been added to the manuscript. We apologize to the reviewers for forgetting to include them in our first submission (they were in a separate file that was never uploaded).

2- The model that T790M positive cells and T790M negative resistant cells differ in their response to AURKB inhibition is based on only a few cell lines. It should be possible to compare Barasertib SA- β -gal vs. PARP cleavage across a larger panel of cell lines. How robust really is this relationship? Also, what is the response of H1975 cells that harbor pre-existing L858R and T790M to Barasertib?

Our reply

Following the reviewer's suggestion, we have analyzed senescence (SA- β -gal) vs. apoptosis (Annexin V) after barasertib treatment in a larger panel of cell lines, namely PC9-GR1, PC9-GR4AZD1, H1975 (T790M positive), PC9-GR3 and PC9-GR1AZD3 (T790M negative). We have also determined the response of H1975 cells to barasertib and S49076. We have found that AURKB inhibition by barasertib induced senescence in all T790M-positive cells but cell death in all T790M negative cells tested. These findings are in agreement with the hypothesis we present in Figure 4i. In addition, we have found that H1975 cells are resistant to barasertib in MTT assays, as expected. We have added the β -gal and Annexin V results in the additional cell lines as new Figures 4f and 4g, and we have included the IC50 and dose-response results for H1975 in the Table 1, Table 2 and Figure 1e of the manuscript. In addition, we have included a comment about the resistance of the H1975 cells to Aurora inhibition in page 7 of the manuscript, and we have modified the results (p.12-13) and discussion (p.19) sections to present the senescence and cell death data in the new cell lines tested. Finally, a sentence has been added in methods section (p.22) about the origin of the H1975 cells used in this study.

3- Similarly, PC-9-GR1 cells are less sensitive to AURKB inhibition compared with the PC9-GR1-AZD (osimertinib-treated) derivative lines. Is this because osimertinib treatment has more fully eradicated T790M clones and there is a greater apoptotic response?

Our reply

The reviewer raises an interesting point. The T790M-positive, osimertinib sensitive PC9-GR1 cells are significantly less sensitive to AURKB inhibition than the osimertinib resistant PC9-GR1-AZD cells, were the p.T790M has disappeared. The reason is exactly what the reviewer points out. As explained above, we have performed cell death and senescence analyses in PC9-GR1 and PC9-GR1AZD3, and we have seen that PC9-GR1AZD3 undergo cell death when treated with AURKB inhibitors, while PC9-GR1 enter senescence (Fig 4f,g).

4- The partial knockdown of AURKB creates a mitotic defect, likely slowing the growth of PC9ER cells - the barasertib drug sensitivity results could be confounded by this. What about the impact of over-expressing of AURKB? Or ideally over-expression of barasertib drug resistant allele?

Our reply

The reviewer expresses his/her concern about the possibility of a longer duplication time in the silenced PC9-ER cells that could confound the results of the IC50 calculation for barasertib. A first line of evidence indicates that the drug sensitivity results of the silenced clones are reliable. Contrary to barasertib, the IC50 of the AURKB-silenced cells for the rest of drugs tested was not altered compared to the controls (see Supplementary Table 1). The effect of a longer duplication time, in case it existed, should be indiscriminate, altering the sensitivity to all agents.

To further excluded the possibility suggested by the reviewer, we have calculated the doubling times of the three clones with partial silencing of AURKB and we have found that they are not significantly different from the parental PC9-ER cells (28-31 h vs. 27 h) and unlikely to confound the results of the proliferation experiments. We have included the doubling time of the silenced clones in the results section of the manuscript (p.10 and Supplementary Table 1), and we have added a sentence with the methodology used in the methods section (p.22).

Based on these results, we have not considered a priority to over-express AURKB and we have concentrated our efforts on CRISPR and additional animal models (see below)

Minor points

1- Supplementary Fig S6A is mislabeled as tumor weight instead of change in tumor volume

Our reply

We thank the reviewer for the observation. The Y label of Supplementary Fig S6A (Fig S9A in the new version of the article) has been corrected

Reviewer #2

1- Figure legends are missing from the main figures, which makes it difficult to fully interpret them

Our reply

Figure legends have been included in the manuscript. Again, we apologize to the reviewers for forgetting to upload them when we first submitted it.

2- In Fig.1D, GAS6 staining pattern is different in each cell line. For PC9-GR1-AZD2 and PC9-GR4-AZD1, most cells are cytoplasm staining. But for other lines, most cells are nuclear staining. As we know, GAS6 should be cytoplasm staining.

Our reply

GAS6 staining pattern was cytoplasmic in all cell lines. However, the low magnification of the images in Fig 1D, the small size and round shape of some of the cell lines and the high intensity of the staining give the false impression that GAS6 is sometimes nuclear. We have added a Supplementary Figure 1c where we show the GAS6 IHC staining of the resistant cells at a higher magnification (x40). In this supplementary figure, which we also mention in the main text of the manuscript (p.5), cytoplasm staining is apparent in all cases, while nuclei appear blue in all the cell lines. We have maintained the lower magnification (x20) in Fig 1d since we think is more appropriate to visualize differences in the levels of protein expression. We have also added a sentence in the material and methods section (p.24) to clarify that all histological images in the paper were obtained with an x20 objective, unless otherwise indicated

3- In Fig.2B, AURKB expression in PC9-GR4 looks stronger than PC9-ER and PC9. For PC9-GR4, every cell has strong staining, but for PC9-ER and PC9, a lot of cells show much weaker staining. If AURKB is the most important gene in this story, the authors should do western blot to compare AURKB expression in each line. mRNA level showed in Fig.2A can't show us the real AURKB level.

Our reply

As the reviewer points out, possible differences in AURKB expression among the cell lines would be a relevant issue. In order to rule them out, we followed the reviewer's suggestion and performed a Western blot experiment to determine AURKB levels in our cell lines, using tubulin as loading control. We did not find any differences in AURKB protein expression among the cells, which was

relatively low in all cases. We have included this result as a new Fig. 2c, and we have mentioned it in the results section of the manuscript (p.9).

4- The authors compared PC9-ER with PC9-GR4, and also compared 11-18-GR5 with 11-18-GR2. From Fig.2E and Fig.5B, we can know the proliferation rate between PC9-ER and PC9-GR4, as well as the proliferation rate between 11-18-GR5 and 11-18-GR2 are big different. For the IC50 comparison, it's hard to exclude the influence derived from different cell proliferation rate

Our reply

As the reviewer points out, there are differences in Ki67 staining –a marker of proliferating cells- in some of the resistant cell lines (Figs 2f and 5b). These differences had no apparent effect on the sensitivity of the cells to AURKB inhibitors by MTT; i.e. all T790M+ and NRAS+ cells were resistant and showed similar dose-response curves, no matter the intensity of the Ki67 staining; and all negative cells were sensitive and had similar curves and IC50s (Tables 2 and 3). If the different percentage of proliferating cells had an influence on the IC50, the IC50 of lines with the same genetic background but dissimilar Ki67 (i.e. PC9-GR1AZD2 and GR1AZD3) would probably be different.

To further exclude the possibility, pointed out by the reviewer, of the different percentage of proliferating cells influencing the IC50s, we have performed two kinds of experiments. First, we have used MTT assays to determine the dose-response curves and IC50s for barasertib of selected cell lines of our panel at incubation times other than 72 h (24 and 48h). We have found that the T790M+ cells are consistently resistant to AURKB inhibition, independently of incubation times; and the T790M- consistently sensitive. In fact, the dose-response curves for barasertib at 48 and 72 h are extremely similar. We have included these data as a new Supplementary Fig 4 in the manuscript, and we have mentioned them in p.7 of the manuscript. Second, we have calculated the doubling times of all the cell lines used in the study, and we have found that they range from 25 to 34 h in the case of the PC9-derived resistant cells, and from 22 to 27 h for the 11-18 resistant lines. These differences are in all likelihood insufficient to have a significant influence on IC50s at the incubation times used. We have included doubling times in Tables 1 and 3; and we have summarized the methodology used in the methods section (p.22).

5- For Fig.3E, it's hard to believe that PC9-ER with partial AURKB knockdown is resistant to AURKB inhibition. If AURKB is the major effector in the resistance, AURKB partial knockdown plus AURKB inhibitor should also show good effect to inhibit cell viability. As well as, Fig.3C showed AURKB partial

knockdown, but it's the expression level of cell pool. Some cells may have very high AURKB knockdown efficiency, and some other cells may have very low AURKB knockdown efficiency, just like the pattern showed in Fig. 3D. If so, cells with high AURKB knockdown efficiency and cells with low AURKB knockdown efficiency will perform big different. Under this condition, it's better to use CRISPR system and select half knockout clones for the experiment.

Our reply

Fig 3c (Fig 3d in the new version of the manuscript) corresponds to the immunofluorescent staining of pH3 in the PC9-ER clones with partial silencing of the *AURKB* expression. However, due to our mistake at not including the figure legends in our first submission, the reviewer has been misled to think that Fig 3C represents AURKB immunofluorescence. Fig 3C was included in the article to demonstrate that pH3 positivity disappears in apparently resting cells after AURKB silencing, just as it happens when AURKB inhibitors are used.

The misassumption about Fig 3C (now Figure 3d) leads the reviewer to suggest that some cells might have low and others high AURKB knockdown efficiency. Two lines of evidence indicate that this is probably not the case. First, we performed ICC for AURKB in the silenced clones, and we did not observe significant differences in AURKB staining among the cells. Second, the clones with AURKB silencing presented a remarkably different morphology compared to parental PC9-ER or PC9-ER cells transfected with a control plasmid. In particular, as explained in the results section of the manuscript, they showed an enlarged cytoplasm, frequent aberrant mitotic figures and a significant fraction of polynucleated cells, a phenotype resembling the morphological changes observed when treating cells with AURKB inhibitors. Remarkably, this phenotype was uniformly observed among the partly silenced cells. If the knockdown efficiency was significantly different, it is to be expected that morphological differences would be apparent from cell to cell. We have included in the manuscript microphotographs of the silenced cells as Supplementary Figure 8, which are referred to in the results section of the manuscript (p.10). In addition, as suggested by the reviewer, we have performed CRISPR (see below)

The reviewer also expresses some doubts about the fact that clones with partial AURKB silencing were resistant to barasertib and suggests that, if AURKB is the major effector in the resistance, they should be sensitive to the drug. Silencing has been used in many published studies about targeted agents. When clones where a particular gene has been silenced become specifically resistant to a drug, it is generally considered to support the idea that the corresponding protein is the target (or one of the targets) of the drug. In our case, partly silenced clones became resistant to barasertib, a drug that has a strong specificity for AURKB (0.37 nM in a cell-free assay, ~3700 fold more

selective for Aurora B over Aurora A). In addition, the silenced clones also became resistant to the other two drugs targeting AURKB tested, namely tozasertib and S49076; while the IC50 for drugs with other targets (gefitinib or BGB324) was unchanged (see Supplementary Table 2). Taken together, these findings indicate that partial silencing of AURKB expression leads to resistance to AURKB inhibitors, as expected. Silenced cells probably develop alternative pathways to circumvent AURKB and, in consequence, become more resistant to AURKB blockade. To further support this point, we have followed the reviewer's advice and we have used CRISPR to silence AURKB in the PC9-ER cells. Similarly to the results obtained when using shRNA, only clones with partial silencing were isolated. The AURKB silenced clones showed a >10-fold increase in the IC50 for barasertib, while the IC50s for gefitinib remained unchanged. The data obtained in the CRISPR experiments have been added as new Supplementary Fig 8b and 8c, and they have been explained in the results section of the manuscript, p11.

6- For Fig. 3F, the authors only showed in vivo S49076 treatment for one cell line. They should compare PC9-ER with PC9-GR4 tumors. If PC9-ER tumors are more sensitive to AURKB inhibition than PC9-GR4, which will help them to support their conclusion.

Our reply

As suggested by the reviewer, we have tested S49076 in PC9-GR4 tumors growing in nude mice. We selected the highest concentration assayed in PC9-ER, 37.5 mg S49076/kg/day, and we found that the PC9-GR4 xenografts are less sensitive to the compound than the PC9-ER xenografts. At the end of the 3-week treatment, there was a 55% reduction in tumor growth in the case of PC9-ER, with an average 4.0 fold increase in the volume of the control xenografts treated with vehicle vs. 1.8-fold in the treated tumors. In contrast, the inhibition was only 21% in the PC-GR4 xenografts (7.0 vs. 5.5-fold increase in tumor volume). We have included the results of the experiment in PC9-GR4 xenografts in Fig 3f and Supplementary Fig 9 and we have added a sentence in the results section (p.11)

7- After AURKB inhibition, T790M resistant cells will finally go senescence, and non-T790M resistant cells will go apoptosis. Cells go senescence, they can't proliferate although they may be still alive, so the final tumor inhibition effect by AURKB inhibition may be very similar. The in vitro experiment only lasts for a few days, and we can't see the long term effect. The authors should do more in vivo treatment study to support their conclusion.

Our reply

As explained above, we have done more in vivo studies, testing S49076 in PC9-GR4 xenografts.

8- In Fig.S4D, PC9-ER (direct count) and PC9-GR4 (direct count) didn't show different sensitivity to Barasertib. In Fig.S7B, 11-8 GR2 (number of cells) and 11-18 GR5 (number of cells) didn't show different sensitivity to Barasertib. These results are not consistent with the author's description in the manuscript.

Our reply

The results of Fig S4D and Fig S7B are consistent with the description in the manuscript. However, the way in which data are plotted can give the false impression that they are not, as commented by the reviewer. For this reason, we have added bar plots with the final MTT and direct cell counts obtained for PC9-ER vs. PC9-GR4 (new Supplementary Fig 7c) and 11-18-GR2 vs. GR5 (new Supplementary Fig 10c) to the results presented in old Fig S4 and Fig S7 (now figures 7 and 10). These bar plots correspond exactly to the same data presented in the 72h timepoint in the growth curves and demonstrate that barasertib is significantly more active in cells without resistance mutations also when direct counting is used instead of MTT.

9.- In Table 2, PC9-GR2 are T790M negative, but it's not sensitive to Barasertib; PC9-GR4-AZD2 only have 0.03% T790M allelic fraction which is very close to negative, but it's also not sensitive to Barasertib. These two cases are not consistent with the author's points. Besides AURKB activation, any other mechanisms involved in this process?

Our reply

Among the 10 cell lines of our panel negative for T790M and NRAS mutations, only PC9-GR2 was found to be insensitive to barasertib. In addition, as the reviewer points out, the allelic fraction of the T790M mutation in the PC9-GR4-AZD2 seems insufficient to explain the insensitivity to Barasertib. The PC9-GR2 line shows activation of MET, while the PC9-GR4-AZD2 line over-expresses FGFR1 (see Fig 1). In these two cases, the intensity of the oncogenic stimuli provided for MET activation and FGFR1 up-regulation could be higher than in other cell lines; enough to drive not a minor but a significant fraction of the cell populations to senescence instead of apoptosis, when treated with AURKB inhibitors (see Discussion, page 19). Alternatively, additional, undetected mechanisms of resistance present exclusively in PC9-GR2 and PC9-GR4-AZD2 could

contribute to the oncogenic stimuli and/or the lack of sensitivity to AURKB inhibition. NGS of the two cell lines with a 20-gene panel did not reveal additional mutations, but PC9-GR2 was relatively sensitive to the AURKA/AURKB inhibitor tozasertib. Therefore, in this particular cell line, we cannot discard a role for AURKA activation, which has been described as a mechanism associated with resistance to EGFR TKIs in an article published when our work was under review (Khyati et al, Nat Med 2018, 10.1038/s41591-018-0264-7, published online on November 26th). Another possibility is the presence in these two cell lines of a RB wt genotype, which has also been demonstrated to confer resistance to AURKB inhibition in another article published on line when our work was under review (Oser et al, Cancer Disc 2018, doi: 10.1158/2159-8290.CD-18-0389, published online on October 29th). Unfortunately, the NGS panel we used for our cells did not include *RB1*.

To gain further insight into this question, we performed a β -gal analysis of the two cell lines, revealing that barasertib induces senescence in both cases; the microphotographs obtained are presented below. This finding supports the above-mentioned hypothesis of a “strong” oncogenic stimuli in both cell lines.

10.- For 11-18 derived resistant cells, the emergence of Q61 NRAS mutations in 11-18 GR1, GR2, GR4, GR6 is 4-22% allelic fractions, which means most cells in each line don't have NRAS mutation, so it's hard to understand these very small population mediates the resistance to AURKB inhibition in IC50 assay which only lasts for a few days. The same concern is also exists in PC-9 based experiments.

Our reply

The reviewer raises a relevant question. In order to find an explanation, we re-analyzed the NGS data of the cell lines included in the article in order to detect not only mutations but also copy number variations (CNVs). We found that the parental and the resistant 11-18 lines have a 4 or 5-fold amplification of the *NRAS* gene, meaning 8 to 10 copies of the gene per cell. This number of copies is theoretically sufficient to account for at least 1 copy of the mutant allele per cell in 11-18 GR1, 11-18 GR2 and 11-18 GR6; which have allelic fractions of 9-22%. The only exception are the 11-18 GR4 cells, with a 5-fold amplification of *NRAS* and a 4% allelic fraction of the Q61L mutation, meaning that the percentage of cells with the *NRAS* mutation might represent up to 50%. This is a significant population, and we think it could be sufficient to mediate resistance to barasertib. Since we think these are relevant results, we have added the *NRAS* amplification data to Table 3, and we have included a sentence in the results section (p.14).

Regarding the PC9-derived cells, the same reasoning could be applied. The *EGFR* gene has been reported to be amplified in the PC9 cell lines, and our re-analysis of the NGS data revealed a 3 to 4-fold amplification of EGFR in all the parental and resistant PC9 cells. In view of this amplification, the allelic fractions of the T790M in PC9-GR1 and PC9-GR4 (25 and 38%, we have included them in p.5 of the manuscript) are more than sufficient to account for ≥ 1 copy of mutant allele per cell.

REVIEWERS' COMMENTS:

Reviewer #1 (Remarks to the Author):

The authors have satisfactorily addressed my concerns

Reviewer #2 (Remarks to the Author):

the authors have addressed all my comments adequately.